

# Variational inference for approximate objective priors using neural networks

**COMPUTO**

**ISSN 2824-7795**

Nils Baillie[1]    Université Paris-Saclay, CEA, Service d'Études Mécaniques et Thermiques, 91191 Gif-sur-Yvette, France

CMAP, CNRS, École polytechnique, Institut Polytechnique de Paris, 91120 Palaiseau, France

Antoine Van Biesbroeck    Université Paris-Saclay, CEA, Service d'Études Mécaniques et Thermiques, 91191 Gif-sur-Yvette, France

CMAP, CNRS, École polytechnique, Institut Polytechnique de Paris, 91120 Palaiseau, France

Clément Gauchy    Université Paris-Saclay, CEA, Service de Génie Logiciel pour la Simulation, 91191 Gif-sur-Yvette, France

Date published: 2025-08-26    Last modified: 2025-08-26

### Abstract

In Bayesian statistics, the choice of the prior can have an important influence on the posterior and the parameter estimation, especially when few data samples are available. To limit the added subjectivity from a priori information, one can use the framework of objective priors, more particularly, we focus on reference priors in this work. However, computing such priors is a difficult task in general. Hence, we consider cases where the reference prior simplifies to the Jeffreys prior. We develop in this paper a flexible algorithm based on variational inference which computes approximations of priors from a set of parametric distributions using neural networks. We also show that our algorithm can retrieve modified Jeffreys priors when constraints are specified in the optimization problem to ensure the solution is proper. We propose a simple method to recover a relevant approximation of the parametric posterior distribution using Markov Chain Monte Carlo (MCMC) methods even if density function of the parametric prior is not known in general. Numerical experiments on several statistical models of increasing complexity are presented. We show the usefulness of this approach by recovering the target distribution. The performance of the algorithm is evaluated on both prior and posterior distributions, jointly using variational inference and MCMC sampling.

*Keywords:* Reference priors, Variational inference, Neural networks

# Contents

[1]Corresponding author: nils.baillie@cea.fr

# 1   Introduction

The Bayesian approach to statistical inference aims to produce estimations using the posterior distribution. The latter is derived by updating the prior distribution with the observed statistics thanks to Bayes' theorem. However, the shape of the posterior can be heavily influenced by the prior choice when the amount of available data is limited or when the prior distribution is highly informative. For this reason, selecting a prior remains a daunting task that must be handled carefully. Hence, systematic methods have been introduced by statisticians to help in the choice of the prior distribution, both in cases where subjective knowledge is available or not (Press (2009)). Kass and Wasserman (1996) propose different ways of defining the level of non-informativeness of a prior distribution. The most famous method is the Maximum Entropy (ME) prior distribution that has been popularized by Jaynes (1957). In a Bayesian context, ME and Maximal Data Information (MDI) priors have been studied by Zellner (1996), Soofi (2000). Other candidates for objective priors are the so-called matching priors (Reid, Mukerjee, and Fraser (2003)), which are priors such that the Bayesian posterior credible intervals correspond to confidence intervals of the sampling model. Moreover, when a simpler model is known, the Penalizing Complexity (PC) priors are yet another rationale of choosing an objective (or reference) prior distribution (Simpson et al. (2017)).

In this paper, we will focus on the reference prior theory. First introduced in Bernardo (1979a) and further formalized in Berger, Bernardo, and Sun (2009), the main rationale behind the reference prior theory is the maximization of the information brought by the data during Bayesian inference. Specifically, reference priors (RPs) constructed to maximize the mutual information metric, which is defined as a divergence between itself and the posterior. In this way, it ensures that the data plays a dominant role in the Bayesian framework. There is consensus that the definition of RPs in high dimensions should be more subtle than simply maximizing the mutual information (see e.g. Berger, Bernardo, and Sun (2015)). A common approach consists in a hierarchical construction of reference priors, firstly mentioned in Bernardo (1979b) and detailed further in Berger and Bernardo (1992b). In this approach, an ordering is imposed on groups of parameters, and the reference prior is derived by sequentially maximizing the mutual information for each group.

Reference priors are used in various statistical models, such as Gaussian process-based models (Paulo (2005), Gu and Berger (2016)), generalized linear models (Natarajan and Kass (2000)), and even Bayesian Neural Networks (Gao, Ramesh, and Chaudhari (2022)). The RPs are recognized for their objective nature in practical studies (D'Andrea (2021), Li and Gu (2021), Van Biesbroeck et al. (2024)), yet they suffer from their low computational feasibility. Indeed, the expression of the RPs often leads to an intricate theoretical expression, which necessitates a heavy numerical cost to be derived that

becomes even more cumbersome as the dimensionality of the problem increases. Moreover, in many applications, a posteriori estimates are obtained using Markov Chain Monte Carlo (MCMC) methods, which require a large number of prior evaluations, further compounding the computational burden. The hierarchical construction of reference priors aggravates this problem even more, for that reason, we will focus solely on the maximization of the mutual information, which corresponds to the special case where no ordering is imposed on the parameters. In this context, it has been shown by Clarke and Barron (1994), and more recently by Van Biesbroeck (2024a) in a more general case, that the Jeffreys prior (Jeffreys (1946)) is the prior that maximizes the mutual information when the number of data samples tends to infinity. Hence, it will serve as the target distribution in our applications.

In general, when we look for sampling or approximating a probability distribution, several approaches arise and may be used within a Bayesian framework. In this work, we focus on variational inference methods. Variational inference seeks to approximate a complex target distribution $p$, —e.g. a posterior— by optimizing over a family of simpler parameterized distributions $q_\lambda$. The goal then is to find the distribution $q_{\lambda*}$ that is the best approximation of $p$ by minimizing a divergence, such as the Kullback-Leibler (KL) divergence. Variational inference methods have been widely adopted in various contexts, including popular models such as Variational Autoencoders (VAEs) (Kingma and Welling (2019)), which are a class of generative models where one wants to learn the underlying distribution of data samples. We can also mention normalizing flows (Papamakarios et al. (2021), Kobyzev, Prince, and Brubaker (2021)), which consider diffeomorphism transformations to recover the density of the approximated distribution from the simpler one taken as input.

Variational inference seems especially relevant in a context where one wants to approximate prior distributions defined as maximizers of a given metric. This kind of approach was introduced in Nalisnick and Smyth (2017) and Gauchy et al. (2023) in order to approximate the Jeffreys prior in one-dimensional models. The main difference being the choice of the objective function. In Nalisnick and Smyth (2017), the authors propose a variational inference procedure using a lower bound of the mutual information as an optimization criterion, whereas in Gauchy et al. (2023), stochastic gradient ascent is directly applied on the mutual information criterion.

By building on these foundations, this paper proposes a novel variational inference algorithm designed to approximate reference priors by maximizing mutual information. Specifically, we focus on the case where no ordering is imposed on the parameters, in which case the reference prior coincides with the Jeffreys prior. For simplicity, we refer to them as variational approximations of the reference priors (VA-RPs).

As in Nalisnick and Smyth (2017) and Gauchy et al. (2023), the Jeffreys prior is approximated in a parametric family of probability distributions implicitly defined by the push-forward probability distribution through a nonlinear function (see e.g. Papamakarios et al. (2021) and Marzouk et al. (2016)). We will focus in this paper to push-forward probability measures through neural networks. In comparison with the previous works, we benchmark extensively our algorithm on statistical models of different complexity and nature to assess its robustness. We also extend our algorithm to handle a more general case where a generalized mutual information criterion is defined using $f$-divergences (Van Biesbroeck (2024a)). In this paper, we restrict the different benchmarks to $\alpha$-divergences. Additionally, we extend the framework to allow the integration of linear constraints on the prior in the pipeline. That last feature permits handling situations where the Jeffreys prior may be improper (i.e. it integrates to infinity). Improper priors pose a challenge because (i) one can not sample from the a priori distribution, and (ii) they do not ensure that the posterior is proper, jeopardizing a posteriori inference. Recent work detailed in Van Biesbroeck (2024b) introduces linear constraints that ensure the proper aspects of priors maximizing the mutual information. Our algorithm incorporates these constraints, providing a principled way to address improper priors and ensuring that the resulting posterior distributions are well-defined and suitable for practical use.

First, we will introduce the reference prior theory of Bernardo (1979b) and the recent developments around generalized reference priors made by Van Biesbroeck (2024a) in Section 2. Next, the methodology to construct VA-RPs is detailed in Section 3. A stochastic gradient algorithm is proposed, as well as an augmented Lagrangian algorithm for the constrained optimization problem, for learning the parameters of an implicitly defined probability density function that will approximate the target prior. Moreover, a mindful trick to sample from the posterior distribution by MCMC using the implicitly defined prior distribution is proposed. In Section 4, different numerical experiments from various test cases are carried out in order to benchmark the VA-RP. Analytical statistical models where the Jeffreys prior is known are tested to allow comparison between the VA-RP and the Jeffreys prior.

## 2 Reference priors theory

The reference prior theory fits into the usual framework of statistical inference. The situation is the following: we observe i.i.d data samples $\mathbf{X} = (X_1, ..., X_N) \in \mathcal{X}^N$ with $\mathcal{X} \subset \mathbb{R}^d$. We suppose that the likelihood function $L_N(\mathbf{X}|\theta) = \prod_{i=1}^N L(X_i|\theta)$ is known and $\theta \in \Theta \subset \mathbb{R}^q$ is the parameter we want to infer. Since we use the Bayesian framework, $\theta$ is considered to be a random variable with a prior distribution $\pi$. We also define the marginal likelihood $p_{\pi,N}(\mathbf{X}) = \int_\Theta \pi(\theta) L_N(\mathbf{X}|\theta) d\theta$ associated to the marginal probability measure $\mathbb{P}_{\pi,N}$. The non-asymptotic RP, first introduced in Bernardo (1979a) and formalized in Berger, Bernardo, and Sun (2009), is defined to be one of the priors verifying:

$$\pi^* \in \underset{\pi \in \mathcal{P}}{\operatorname{argmax}} I(\pi; L_N), \tag{1}$$

where $\mathcal{P}$ is a class of admissible probability distributions and $I(\pi; L_N)$ is the mutual information for the prior $\pi$ and the likelihood $L_N$ between the random variable of the parameters $\theta \sim \pi$ and the random variable of the data $\mathbf{X} \sim \mathbb{P}_{\pi,N}$:

$$I(\pi; L_N) = \int_{\mathcal{X}^N} \mathrm{KL}(\pi(\cdot|\mathbf{X}) \| \pi) p_{\pi,N}(\mathbf{X}) d\mathbf{X} \tag{2}$$

Hence, $\pi^*$ is a prior that maximizes the Kullback-Leibler divergence between itself and its posterior averaged by the marginal distribution of datasets. The Kullback-Leibler divergence between two probability measures of density $p$ and $q$ defined on a generic set $\Omega$ writes:

$$\mathrm{KL}(p \| q) = \int_\Omega \log\left(\frac{p(\omega)}{q(\omega)}\right) p(\omega) d\omega.$$

Thus, $\pi^*$ is the prior that maximizes the influence of the data on the posterior distribution, justifying its reference (or objective) properties. The prior $\pi^*$ can also be interpreted using channel coding and information theory (MacKay (2003), chapter 9). Indeed, remark that $I(\pi; L_N)$ corresponds to the mutual information $I(\theta, \mathbf{X})$ between the random variable $\theta \sim \pi$ and the data $\mathbf{X} \sim \mathbb{P}_{\pi,N}$, it measures the information that conveys the data $\mathbf{X}$ about the parameters $\theta$. The maximal value of this mutual information is defined as the channel's capacity. $\pi^*$ thus corresponds to the prior distribution that maximizes the information about $\theta$ conveyed by the data $\mathbf{X}$.

Using Fubini's theorem and Bayes' theorem, we can derive an alternative and more practical expression for the mutual information:

$$I(\pi; L_N) = \int_\Theta \mathrm{KL}(L_N(\cdot|\theta) \| p_{\pi,N}) \pi(\theta) d\theta. \tag{3}$$

A more generalized definition of mutual information has been proposed in Van Biesbroeck (2024a) using $f$-divergences. The $f$-divergence mutual information is defined by

$$I_{\mathrm{D}_f}(\pi; L_N) = \int_\Theta \mathrm{D}_f(p_{\pi,N} \| L_N(\cdot|\theta)) \pi(\theta) d\theta, \tag{4}$$

with

$$\mathrm{D}_f(p \,\|\, q) = \int_\Omega f\left(\frac{p(\omega)}{q(\omega)}\right) q(\omega) d\omega,$$

where $f$ is usually chosen to be a convex function mapping 1 to 0. Remark that the classical mutual
information is obtained by choosing $f = -\log$, indeed, $\mathrm{D}_{-\log}(p \,\|\, q) = \mathrm{KL}(q \,\|\, p)$. The formal RP is
defined as $N$ goes to infinity, but in practice we are restricted to the case where $N$ takes a finite value.
However, the limit case $N \to +\infty$ is relevant because it has been shown in Clarke and Barron (1994),
Van Biesbroeck (2024a) that the solution of this asymptotic problem is the Jeffreys prior when the
mutual information is expressed as in Equation 2, or when it is defined using an $\alpha$-divergence, as in
Equation 4 with $f = f_\alpha$, where:

$$f_\alpha(x) = \frac{x^\alpha - \alpha x - (1 - \alpha)}{\alpha(\alpha - 1)}, \quad \alpha \in (0, 1). \tag{5}$$

The Jeffreys prior, denoted by $J$, is defined as follows:

$$J(\theta) \propto \det(\mathscr{I}(\theta))^{1/2} \quad \text{with} \quad \mathscr{I}(\theta) = -\int_{\mathcal{X}^N} \frac{\partial^2 \log L_N}{\partial \theta^2}(\mathbf{X} \,|\, \theta) \cdot L_N(\mathbf{X} \,|\, \theta) \, d\mathbf{X}.$$

We suppose that the likelihood function is smooth such that the Fisher information matrix $\mathscr{I}$ is well-
defined. The Jeffreys prior and the RP have the relevant property to be "invariant by reparametriza-
tion":

$$\forall \varphi \text{ diffeomorphism}, \quad J(\theta) = \left|\frac{\partial \varphi}{\partial \theta}\right| \cdot J(\varphi(\theta)).$$

This property expresses non-information in the sense that if there is no information on $\theta$, there
should not be more information on $\varphi(\theta)$ when $\varphi$ is a diffeomorphism: an invertible and differentiable
transformation.

Actually, the historical definition of RPs involves the KL-divergence in the definition of the mutual
information. Yet the use of $\alpha$-divergences instead is relevant because they can be seen as a continuous
path between the KL-divergence and the Reverse-KL-divergence when $\alpha$ varies from 0 to 1. We can
also mention that for $\alpha = 1/2$, the $\alpha$-divergence is the squared Hellinger distance whose square root
is a metric since it is symmetric and verifies the triangle inequality.

Technically, the formal RP is constructed such that its projection on every compact subset (or open
subset in Muré (2018)) of $\Theta$ maximizes asymptotically the mutual information, which allows for
improper distributions to be RPs in some cases. The Jeffreys prior is itself often improper.

In our algorithm we consider probability distributions defined on the space $\Theta$ and not on sequences
of subsets. A consequence of this statement is that our algorithm may tend to approximate improper
priors in some cases. Indeed, any given sample by our algorithm results, by construction, from a
proper distribution, which is expected to be a good approximation of the solution of the optimization
problem expressed in Equation 1. This approach is justified to some extent since in the context of
Q-vague convergence defined in Bioche and Druilhet (2016) for instance, improper priors can be
the limit of sequences of proper priors. Although this theoretical notion of convergence is defined,
no concrete metric is given, making quantification of the difference between proper and improper
priors infeasible in practice.

The term "reference prior" is now associated with a more general, hierarchical construction. We
mentioned in the introduction the hierarchical construction of the reference prior, we present rapidly
the case where the dimension $q = 2$, i.e. $\theta = (\theta_1, \theta_2) \in \Theta_1 \times \Theta_2$ with $\theta_1$ and $\theta_2$ being in their own
separate groups:

- We obtain the first level conditional prior: $\pi_1^*(\cdot \,|\, \theta_2)$ on $\theta_1$ by maximizing asymptotically the mutual information with fixed $\theta_2$ in the likelihood $L_N$.
- We define the second level likelihood using the previous prior as follows:

$$L_N'(X \,|\, \theta_2) = \int_{\Theta_1} L_N(X \,|\, \theta_1, \theta_2) \pi_1^*(\theta_1 \,|\, \theta_2) d\theta_1.$$

- We define and solve the corresponding asymptotic optimization problem with this function as our main likelihood function so we can obtain the second level prior: $\pi_2^*$ on $\theta_2$.
- This defines the hierarchical RP on $\theta$, which is of the form: $\pi^*(\theta) = \pi_1^*(\theta_1 \,|\, \theta_2) \pi_2^*(\theta_2)$.

This construction can be extended to any number of groups of parameters with any ordering as presented in Berger and Bernardo (1992b). However, it is important to note that priors defined through this procedure can still be improper.

In summary, we introduced several priors: the Jeffreys prior, the non-asymptotic RP that maximizes the generalized mutual information, which depends on the chosen $f$-divergence and the value of $N$, the formal RP, that is obtained such that its projection on every (compact) subset maximizes asymptotically the generalized mutual information, hence it only depends on the $f$-divergence, and finally, the reference prior in the hierarchical sense. The latter reduces to the formal RP (i) in the one-dimensional case and (ii) in the multi-dimensional case, when all components of $\theta$ are placed in the same group. We will always be in one of these two cases in the following. In very specific situations, where the likelihood function is non-regular (Ghosal and Samanta (1997)) or because of the choice of $f$ (Liu et al. (2014)), the formal RP and the Jeffreys prior can be different. However, as long as the likelihood is smooth which is verified for most statistical models and the KL-divergence or the $\alpha$-divergence with $\alpha \in (0, 1)$ is used, these two priors are actually the same.

The algorithm we develop aims at solving the mutual information optimization problem with $N$ fixed, thus our target prior is technically the non-asymptotic RP, nevertheless, the latter has no closed form expression, making the validation of the algorithm infeasible. If $N$ is large enough, this prior should be close to the formal RP which is equal to the Jeffreys prior in this framework. Hence, the Jeffreys prior serves as the target prior in the numerical applications because it can either be computed explicitly or approximated through numerical integration.

Furthermore, as mentioned in the introduction, improper priors can also compromise the validity of a posteriori estimates in some cases. To address this issue, we adapted our algorithm to handle the developments made in Van Biesbroeck (2024b), which suggest a method to define proper objective priors by simply resolving a constrained version of the initial optimization problem:

$$\tilde{\pi}^* \in \operatorname*{argmax}_{\substack{\pi \text{ prior} \\ \text{s.t. } \mathscr{C}(\pi) < \infty}} I_{D_{f_\alpha}}(\pi; L_N), \tag{6}$$

where $\mathscr{C}(\pi)$ defines a constraint of the form $\int_\Theta a(\theta)\pi(\theta)d\theta$, $a$ being a positive function. When the mutual information in the above optimization problem is defined from an $\alpha$-divergence, and when $a$ verifies

$$\int_\Theta J(\theta)a(\theta)^{1/\alpha}d\theta < \infty \quad \text{and} \quad \int_\Theta J(\theta)a(\theta)^{1+1/\alpha}d\theta < \infty, \tag{7}$$

the author has proven that the constrained solution $\tilde{\pi}^*$ asymptotically takes the following form:

$$\tilde{\pi}^*(\theta) \propto J(\theta)a(\theta)^{1/\alpha},$$

which is proper. This result implies that in the case where constraints are imposed, the target prior becomes this modified version of the Jeffreys prior.

## 3 Variational approximation of the reference prior (VA-RP)

### 3.1 Implicitly defined parametric probability distributions using neural networks

Variational inference refers to techniques that aim to approximate a probability distribution by solving an optimization problem —that often takes a variational form, such as maximizing evidence lower bound (ELBO) (Kingma and Welling (2014)). It is thus relevant to consider them for approximating RPs, as the goal is to maximize, w.r.t. the prior, the mutual information defined in Equation 3.

We restrict the set of priors to a parametric space $\{\pi_\lambda, \lambda \in \Lambda\}$, $\Lambda \subset \mathbb{R}^L$, reducing the original optimization problem into a finite-dimensional one. The optimization problem in Equation 1 or Equation 6 becomes finding $\underset{\lambda \in \Lambda}{\arg\max}\, I_{D_f}(\pi_\lambda; L_N)$. Our approach is to define the set of priors $\pi_\lambda$ implicitly, as in Gauchy et al. (2023):

$$\theta \sim \pi_\lambda \iff \theta = g(\lambda, \varepsilon) \quad \text{and} \quad \varepsilon \sim \mathbb{P}_\varepsilon.$$

Here, $g$ is a measurable function parameterized by $\lambda$, typically a neural network with $\lambda$ corresponding to its weights and biases, and we impose that $g$ is differentiable with respect to $\lambda$. The variable $\varepsilon$ can be seen as a latent variable. It has an easy-to-sample distribution $\mathbb{P}_\varepsilon$ with a simple density function. In practice we use the centered multivariate Gaussian $\mathcal{N}(0, \mathbb{I}_{p\times p})$. The construction described above allows the consideration of a vast family of priors. However, except in very simple cases, the density of $\pi_\lambda$ is not known and cannot be evaluated. Only samples of $\theta \sim \pi_\lambda$ can be obtained.

In the work of Nalisnick and Smyth (2017), this implicit sampling method is compared to several other algorithms used to learn RPs in the case of one-dimensional models, where the RP is always the Jeffreys prior. Among these methods, we can mention an algorithm proposed by Berger, Bernardo, and Sun (2009) which does not sample from the RP but only evaluates it for specific points, or an MCMC-based approach by Lafferty and Wasserman (2001), which is inspired from the previous one but can sample from the RP.

According to this comparison, implicit sampling is, in the worst case, competitive with the other methods, but achieves state-of-the-art results in the best case. Hence, computing the variational approximation of the RP, which we will refer to as the VA-RP, seems to be a promising technique. We admit that the term VA-RP is a slight abuse of terminology in our case since (i) the target prior is the (eventually constrained) Jeffreys prior, which is not necessarily the reference prior when an ordering is imposed on the parameters; and (ii) there is no guarantee that this target prior can be actually reproduced by the neural network. Indeed, the VA-RP tends to be the prior that maximizes the mutual information for a fixed value of $N$, within a family of priors that is, by design, parameterized by $\lambda$. Since we are aware of those approximations, we strive to assess that our priors are good approximations of the target priors in our numerical experiments.

The situations presented by Gauchy et al. (2023) and Nalisnick and Smyth (2017) are in dimension one and use the Kullback-Leibler divergence within the definition of the mutual information.

The construction of the algorithm that we propose in the following accommodates multi-dimensional modeling. It is also compatible with the extended form of the mutual information, as defined in Equation 3 from an $f$-divergence.

The choice of the neural network is up to the user, we will showcase in our numerical applications mostly simple networks, composed of one fully connected linear layer and one activation function. However, the method can be used with deeper networks, such as normalizing flows (Papamakarios et al. (2021)), or larger networks obtained through a mixture model of smaller networks utilizing the "Gumbel-Softmax trick" (Jang, Gu, and Poole (2017)) for example. Such choices lead to more flexible parametric distributions, but increase the difficulty of fine-tuning hyperparameters.

## 3.2 Learning the VA-RP using stochastic gradient algorithm

The VA-RP is formulated as the solution to the following optimization problem:

$$\pi_{\lambda^*} = \underset{\lambda \in \Lambda}{\operatorname{argmax}} \, \mathscr{O}_{\mathrm{D}_f}(\pi_\lambda; L_N), \tag{8}$$

where $\pi_\lambda$ is parameterized through the relation between a latent variable $\varepsilon$ and the parameter $\theta$, as outlined in the preceding Section. The function $\mathscr{O}_{\mathrm{D}_f}$ is called the objective function, it is maximized using stochastic gradient optimization, following the approach described in Gauchy et al. (2023).

It is intuitive to fix $\mathscr{O}_{\mathrm{D}_f}$ to equal $I_{\mathrm{D}_f}$, in order to maximize the mutual information of interest. In this Section, we suggest alternative objective functions that can be considered to compute the VA-RP. Our method is adaptable to any objective function $\mathscr{O}_{\mathrm{D}_f}$ that satisfies the following definition.

**Definition 1.** An objective function $\mathscr{O}_{\mathrm{D}_f} : \lambda \in \Lambda \mapsto \mathscr{O}_{\mathrm{D}_f}(\pi_\lambda; L_N) \in \mathbb{R}$ is said to be admissible if there exists a mapping $\tilde{\mathscr{O}}_{\mathrm{D}_f} : \Theta \to \mathbb{R}$ such that the gradient of $\mathscr{O}_{\mathrm{D}_f}$ w.r.t. $\lambda = (\lambda_1, \dots, \lambda_L)$ is

$$\frac{\partial \mathscr{O}_{\mathrm{D}_f}}{\partial \lambda_l}(\pi_\lambda; L_N) = \mathbb{E}_\varepsilon \left[ \sum_{j=1}^q \frac{\partial \tilde{\mathscr{O}}_{\mathrm{D}_f}}{\partial \theta_j}(g(\lambda, \varepsilon)) \frac{\partial g_j}{\partial \lambda_l}(\lambda, \varepsilon) \right] \tag{9}$$

for any $l \in \{1, \dots, L\}$.

Here, $\tilde{\mathscr{O}}_{\mathrm{D}_f}$ is a generic notation for a function that depends in practice on $f$ and the likelihood function. We also assume that its gradient is computed using Monte Carlo sampling. The framework of admissible objective functions allows for flexible implementation, as it permits the separation of sampling and differentiation operations:

- The gradient of $\tilde{\mathscr{O}}_{\mathrm{D}_f}$ mostly relies on random sampling and depends only on the likelihood $L_N$ and the function $f$.
- The gradient of $g$ is computed independently. In practice, it is possible to leverage usual differentiation techniques for the neural network. In our work, we rely on PyTorch's automatic differentiation feature "autograd" (Paszke et al. (2019)).

This separation is advantageous as automatic differentiation tools —such as autograd— are well-suited to differentiating complex networks but struggle with functions incorporating randomness.

This way, the optimization problem can be addressed using stochastic gradient optimization, approximating at each step the gradient in Equation 9 via Monte Carlo estimates. In our experiments, the implementation of the algorithm is done with the popular Adam optimizer (Kingma and Ba (2017)), with its default hyperparameters, $\beta_1 = 0.9$ and $\beta_2 = 0.999$. The learning rate is tuned more specifically for each numerical benchmark.

Concerning the choice of objective function, we verify that in appendix Section 6.1

$$\frac{\partial I_{\mathrm{D}_f}}{\partial \lambda_l}(\pi_\lambda; L_N) = \mathbb{E}_\varepsilon \left[ \sum_{j=1}^q F_j \cdot \frac{\partial g_j}{\partial \lambda_l}(\lambda, \varepsilon) \right]$$
$$+ \mathbb{E}_{\theta \sim \pi_\lambda} \left[ \mathbb{E}_{\mathbf{X} \sim L_N(\cdot|\theta)} \left[ \frac{1}{L_N(\mathbf{X}|\theta)} \frac{\partial p_\lambda}{\partial \lambda_l}(\mathbf{X}) f'\left( \frac{p_\lambda(\mathbf{X})}{L_N(\mathbf{X}|\theta)} \right) \right] \right], \tag{10}$$

where:

$$F_j = \mathbb{E}_{\mathbf{X} \sim L_N(\cdot|\theta)} \left[ \frac{\partial \log L_N}{\partial \theta_j}(\mathbf{X}|\theta) F\left( \frac{p_\lambda(\mathbf{X})}{L_N(\mathbf{X}|\theta)} \right) \right],$$

with $F(x) = f(x) - x f'(x)$ and $p_\lambda$ is a shortcut notation for $p_{\pi_\lambda, N}$ being the marginal distribution under $\pi_\lambda$.

280 Remark that only the case $f = -\log$ is considered by Gauchy et al. (2023), but it leads to a sim-
281 plification of the gradient since the second term vanishes. Each term in the above equations is
282 approximated as follows:

$$\begin{cases} p_\lambda(\mathbf{X}) = \mathbb{E}_{\theta \sim \pi_\lambda}[L_N(\mathbf{X} \,|\, \theta)] \approx \dfrac{1}{T} \sum_{t=1}^{T} L_N(\mathbf{X} \,|\, g(\lambda, \varepsilon_t)) \quad \text{where} \quad \varepsilon_1, \dots, \varepsilon_T \sim \mathbb{P}_\varepsilon \\[2ex] F_j \approx \dfrac{1}{U} \sum_{u=1}^{U} \dfrac{\partial \log L_N}{\partial \theta_j}(\mathbf{X}^u \,|\, \theta) F\left( \dfrac{p_\lambda(\mathbf{X}^u)}{L_N(\mathbf{X}^u \,|\, \theta)} \right) \quad \text{where} \quad \mathbf{X}^1, \dots, \mathbf{X}^U \sim \mathbb{P}_{\mathbf{X}|\theta}. \end{cases} \quad (11)$$

283 In their work, Nalisnick and Smyth (2017) propose an alternative objective function to optimize, that
284 we call $B_{\mathrm{D}_f}$.

285 This function corresponds to a lower bound of the mutual information. It is derived from an upper
286 bound on the marginal distribution and relies on maximizing the likelihood. Their approach is only
287 presented for $f = -\log$, we generalize the lower bound for any decreasing function $f$:

$$B_{\mathrm{D}_f}(\pi; L_N) = \int_\Theta \int_{\mathscr{X}^N} f\left( \frac{L_N(\mathbf{X} \,|\, \hat{\theta}_{MLE})}{L_N(\mathbf{X} \,|\, \theta)} \right) \pi(\theta) L_N(\mathbf{X} \,|\, \theta) d\mathbf{X} d\theta,$$

288 where $\hat{\theta}_{MLE}$ being the maximum likelihood estimator (MLE). It only depends on the likelihood and
289 not on $\lambda$ which simplifies the gradient computation:

$$\frac{\partial B_{\mathrm{D}_f}}{\partial \lambda_l}(\pi_\lambda; L_N) = \mathbb{E}_\varepsilon \left[ \sum_{j=1}^{q} \frac{\partial \tilde{B}_{\mathrm{D}_f}}{\partial \theta_j}(g(\lambda, \varepsilon)) \frac{\partial g_j}{\partial \lambda_l}(\lambda, \varepsilon) \right],$$

290 where:

$$\frac{\partial \tilde{B}_{\mathrm{D}_f}}{\partial \theta_j}(\theta) = \mathbb{E}_{\mathbf{X} \sim L_N(\cdot \,|\, \theta)} \left[ \frac{\partial \log L_N}{\partial \theta_j}(\mathbf{X} \,|\, \theta) F\left( \frac{L_N(\mathbf{X} \,|\, \hat{\theta}_{MLE})}{L_N(\mathbf{X} \,|\, \theta)} \right) \right].$$

291 Its form corresponds to the one of an admissible objective function (Equation 9), with:

$$\tilde{B}_{\mathrm{D}_f}(\theta) = \int_{\mathscr{X}^N} L_N(\mathbf{X} \,|\, \theta) f\left( \frac{L_N(\mathbf{X} \,|\, \hat{\theta}_{MLE})}{L_N(\mathbf{X} \,|\, \theta)} \right) d\mathbf{X}.$$

292 Given that $p_\lambda(\mathbf{X}) \leq \max_{\theta' \in \Theta} L_N(\mathbf{X} \,|\, \theta') = L_N(\mathbf{X} \,|\, \hat{\theta}_{MLE})$ for all $\lambda$, we have $B_{\mathrm{D}_f}(\pi_\lambda; L_N) \leq I_{\mathrm{D}_f}(\pi_\lambda; L_N)$.

293 Since $f_\alpha$, used in $\alpha$-divergence (Equation 5), is not decreasing, we replace it with $\hat{f}_\alpha$ defined hereafter,
294 because $\mathrm{D}_{f_\alpha} = \mathrm{D}_{\hat{f}_\alpha}$:

$$\hat{f}_\alpha(x) = \frac{x^\alpha - 1}{\alpha(\alpha - 1)} = f_\alpha(x) + \frac{1}{\alpha - 1}(x - 1).$$

295 The use of this function results in a more stable computation overall. Moreover, one argument for
296 the use of $\alpha$-divergences rather that the KL-divergence, is that we have an universal and explicit
297 upper bound on the mutual information:

$$I_{\mathrm{D}_{f_\alpha}}(\pi; L_N) = I_{\mathrm{D}_{\hat{f}_\alpha}}(\pi; L_N) \leq \hat{f}_\alpha(0) = \frac{1}{\alpha(1 - \alpha)}.$$

298 This bound can be an indicator on how well the mutual information is optimized, although there is
299 no guarantee that it can be attained in general.

 The gradient of the objective function $B_{D_f}$ can be approximated via Monte Carlo, in the same manner
 as in Equation 11.

 It requires to compute the MLE, which can also be done using samples of $\varepsilon$:

$$L_N(\mathbf{X} \,|\, \hat{\theta}_{MLE}) \approx \max_{t \in \{1,...,T\}} L_N(\mathbf{X} \,|\, g(\lambda, \varepsilon_t)) \quad \text{where} \quad \varepsilon_1, ..., \varepsilon_T \sim \mathbb{P}_\varepsilon.$$

### 3.3 Adaptation for the constrained VA-RP

Reference priors and Jeffreys priors are often criticized, because they can lead to improper posteriors.
However, the variational optimization problem defined in Equation 8 can be adapted to incorporate
simple constraints on the prior. As mentioned in Section 2, there exist specific constraints that would
make the theoretical solution proper.

This is also a way to incorporate expert knowledge to some extent. We consider $K$ constraints of the
form:

$$\forall k \in \{1, ..., K\}, \ \mathscr{C}_k(\pi_\lambda) = \mathbb{E}_{\theta \sim \pi_\lambda}[a_k(\theta)] - b_k,$$

with $a_k \colon \Theta \mapsto \mathbb{R}^+$ integrable and linearly independent functions, and $b_k \in \mathbb{R}$. We then adapt the
optimization problem in Equation 8 to propose the following constrained optimization problem:

$$\pi^C_{\lambda^*} \in \underset{\lambda \in \Lambda}{\operatorname{argmax}} \ \mathscr{O}_{D_f}(\pi_\lambda; L_N)$$
$$\text{subject to} \quad \forall k \in \{1, ..., K\}, \ \mathscr{C}_k(\pi_\lambda) = 0,$$

where $\pi^C_{\lambda^*}$ is the constrained VA-RP. The optimization problem with the mutual information has an
explicit asymptotic solution for proper priors verifying the previous conditions:

- In the case of the KL-divergence (Bernardo (2005)):

$$\pi^C(\theta) \propto J(\theta) \exp\left(1 + \sum_{k=1}^{K} v_k a_k(\theta)\right).$$

- In the case of $\alpha$-divergences (Van Biesbroeck (2024b)):

$$\pi^C(\theta) \propto J(\theta)\left(1 + \sum_{k=1}^{K} v_k a_k(\theta)\right)^{1/\alpha}.$$

where $v_1, ..., v_K \in \mathbb{R}$ are constants determined by the constraints.

Recent work by Van Biesbroeck (2024b) makes it possible to build a proper objective prior under a
relevant constraint function with $\alpha$-divergence. The theorem considers $a : \Theta \mapsto \mathbb{R}^+$ which verifies
the conditions expressed in Equation 7. Letting $\mathscr{P}_a$ be the set of proper priors $\pi$ on $\Theta$ such that
$\pi \cdot a \in L^1$, the prior $\tilde{\pi}^*$ that maximizes the mutual information under the constraint $\tilde{\pi}^* \in \mathscr{P}_a$ is:

$$\tilde{\pi}^*(\theta) \propto J(\theta)a(\theta)^{1/\alpha}.$$

We propose the following general method to approximate the VA-RP under such constraints:

- Compute the VA-RP $\pi_\lambda \approx J$, in the same manner as for the unconstrained case.

- Estimate the constants $\mathscr{K}$ and $c$ using Monte Carlo samples from the VA-RP, as:

$$\mathscr{K}_\lambda = \int_\Theta \pi_\lambda(\theta) a(\theta)^{1/\alpha} d\theta \approx \int_\Theta J(\theta) a(\theta)^{1/\alpha} d\theta = \mathscr{K},$$

$$c_\lambda = \int_\Theta \pi_\lambda(\theta) a(\theta)^{1+(1/\alpha)} d\theta \approx \int_\Theta J(\theta) a(\theta)^{1+(1/\alpha)} d\theta = c.$$

- Since we have the equality:

$$\mathbb{E}_{\theta \sim \tilde{\pi}^*}[a(\theta)] = \int_\Theta \tilde{\pi}^*(\theta) a(\theta) d\theta = \frac{1}{\mathscr{K}} \int_\Theta J(\theta) a(\theta)^{1+(1/\alpha)} d\theta = \frac{c}{\mathscr{K}},$$

we compute the constrained VA-RP using the constraint: $\mathbb{E}_{\theta \sim \pi_{\lambda'}}[a(\theta)] = c_\lambda / \mathscr{K}_\lambda$ to approximate $\pi_{\lambda'} \approx \tilde{\pi}^*$.

One might use different variational approximations for $\pi_\lambda$ and $\pi_{\lambda'}$ because $J$ and $\tilde{\pi}^*$ could have very different forms depending on the function $a$.

The idea is to solve the constrained optimization problem as an unconstrained problem but with a Lagrangian as the objective function. We take the work of Nocedal and Wright (2006) as support.

We denote $\eta$ the Lagrange multiplier. Instead of using the usual Lagrangian function, Nocedal and Wright (2006) suggest adding a term defined with $\tilde{\eta}$, a vector with positive components which serve as penalization coefficients, and $\eta'$ which can be thought of a prior estimate of $\eta$, although not in a Bayesian sense. The objective is to find a saddle point $(\lambda^*, \eta^*)$ which is a solution of the updated optimization problem:

$$\max_\lambda \left( \min_\eta \mathscr{O}_{\mathrm{D}_f}(\pi_\lambda; L_N) + \sum_{k=1}^K \eta_k \mathscr{C}_k(\pi_\lambda) + \sum_{k=1}^K \frac{1}{2\tilde{\eta}_k}(\eta_k - \eta'_k)^2 \right).$$

One can see that the third term serves as a penalization for large deviations from $\eta'$. The minimization on $\eta$ is feasible because it is a convex quadratic, and we get $\eta = \eta' - \tilde{\eta} \cdot \mathscr{C}(\pi_\lambda)$. Replacing $\eta$ by its expression leads to the resolution of the problem:

$$\max_\lambda \mathscr{O}_{\mathrm{D}_f}(\pi_\lambda; L_N) + \sum_{k=1}^K \eta'_k \mathscr{C}_k(\pi_\lambda) - \sum_{k=1}^K \frac{\tilde{\eta}_k}{2} \mathscr{C}_k(\pi_\lambda)^2.$$

This motivates the definition of the augmented Lagrangian:

$$\mathscr{L}_A(\lambda, \eta, \tilde{\eta}) = \mathscr{O}_{\mathrm{D}_f}(\pi_\lambda; L_N) + \sum_{k=1}^K \eta_k \mathscr{C}_k(\pi_\lambda) - \sum_{k=1}^K \frac{\tilde{\eta}_k}{2} \mathscr{C}_k(\pi_\lambda)^2.$$

Its gradient has a form that is compatible with our algorithm, as depicted in Section 3.2 (see Equation 9):

$$\frac{\partial \mathscr{L}_A}{\partial \lambda}(\lambda, \eta, \tilde{\eta}) = \frac{\partial \mathscr{O}_{\mathrm{D}_f}}{\partial \lambda}(\pi_\lambda; L_N) + \mathbb{E}_\varepsilon \left[ \left( \sum_{k=1}^K \frac{\partial a_k}{\partial \theta}(g(\lambda, \varepsilon))(\eta_k - \tilde{\eta}_k \mathscr{C}_k(\pi_\lambda)) \right) \frac{\partial g}{\partial \lambda}(\lambda, \varepsilon) \right]$$

$$= \mathbb{E}_\varepsilon \left[ \left( \frac{\partial \tilde{\mathscr{O}}}{\partial \theta}(g(\lambda, \varepsilon)) + \sum_{k=1}^K \frac{\partial a_k}{\partial \theta}(g(\lambda, \varepsilon))(\eta_k - \tilde{\eta}_k \mathscr{C}_k(\pi_\lambda)) \right) \frac{\partial g}{\partial \lambda}(\lambda, \varepsilon) \right].$$

In practice, the augmented Lagrangian algorithm is of the form:

$$\begin{cases} \lambda^{t+1} = \underset{\lambda}{\mathrm{argmax}}\, \mathscr{L}_A(\lambda, \eta^t, \tilde{\eta}) \\ \forall k \in \{1, ..., K\}, \eta_k^{t+1} = \eta_k^t - \tilde{\eta}_k \cdot \mathscr{C}_k(\pi_{\lambda^{t+1}}). \end{cases}$$

344 In our implementation, $\eta$ is updated every 100 epochs. The penalty parameter $\tilde{\eta}$ can be interpreted
345 as the learning rate of $\eta$, we use an adaptive scheme inspired by Basir and Senocak (2023) where
346 we check if the largest constraint value $\|\mathscr{C}(\pi_\lambda)\|_\infty$ is higher than a specified threshold $M$ or not. If
347 $\|\mathscr{C}(\pi_\lambda)\|_\infty > M$, we multiply $\tilde{\eta}$ by $v$, otherwise we divide by $v$. We also impose a maximum value $\tilde{\eta}_{max}$.

## 3.4 Posterior sampling using implicitly defined prior distributions

349 Although our main object of study is the prior distribution, one needs to find the posterior distribution
350 given an observed dataset $\mathbf{X}$ in order to do the inference on $\theta$. The posterior is of the form:

$$\pi_\lambda(\theta|\mathbf{X}) = \frac{\pi_\lambda(\theta)L_N(\mathbf{X}|\theta)}{p_\lambda(\mathbf{X})}.$$

351 As discussed in the introduction, one can approximate the posterior distribution when knowing
352 the prior either by using MCMC or variational inference. In both cases, knowing the marginal
353 distribution is not required. Indeed, MCMC samplers inspired by the Metropolis-Hastings algorithm
354 can be applied, even if the posterior distribution is only known up to a multiplicative constant.
355 The same can be said for variational approximation since the ELBO can be expressed without the
356 marginal.

357 The issue here is that the density function $\pi_\lambda(\theta)$ is not explicit and can not be evaluated, except for
358 very simple cases. However, we imposed that the distribution of the variable $\varepsilon$ is simple enough, so
359 one is able to evaluate its density. We propose to use $\varepsilon$ as the variable of interest instead of $\theta$ because
360 it lets us circumvent this issue. In practice, the idea is to reverse the order of operations: instead of
361 sampling $\varepsilon$, then transforming $\varepsilon$ into $\theta$, which defines the prior on $\theta$, and finally sampling posterior
362 samples of $\theta$ given $X$, one can proceed as follows:

363    • Define the posterior distribution on $\varepsilon$:

$$\pi_{\varepsilon,\lambda}(\varepsilon|\mathbf{X}) = \frac{p_\varepsilon(\varepsilon)L_N(\mathbf{X}|g(\lambda,\varepsilon))}{p_\lambda(\mathbf{X})} \ ,$$

364    where $p_\varepsilon$ is the probability density function of $\varepsilon$. $\pi_{\varepsilon,\lambda}(\varepsilon|\mathbf{X})$ is known up to a multiplicative
365    constant since the marginal $p_\lambda$ is intractable in general. It is indeed a probability distribution
366    on $\mathbb{R}^p$ because:

$$p_\lambda(\mathbf{X}) = \int_\Theta \pi_\lambda(\theta)L_N(\mathbf{X}|\theta)d\theta = \int_{\mathbb{R}^p} L_N(\mathbf{X}|g(\lambda,\varepsilon))d\mathbb{P}_\varepsilon$$

367    • Sample posterior $\varepsilon$ samples from the previous distribution, approximated by MCMC or varia-
368    tional inference.

369    • Apply the transformation $\theta = g(\lambda,\varepsilon)$, and one gets posterior $\theta$ samples: $\theta \sim \pi_\lambda(\cdot|\mathbf{X})$.

370 More precisely, we denote for a fixed dataset $\mathbf{X}$:

$$\theta \sim \tilde{\pi}_\lambda(\cdot|\mathbf{X}) \iff \theta = g(\lambda,\varepsilon) \quad \text{with} \quad \varepsilon \sim \pi_{\varepsilon,\lambda}(\cdot|\mathbf{X}).$$

371 The previous approach is valid because $\pi_\lambda(\cdot|\mathbf{X})$ and $\tilde{\pi}_\lambda(\cdot|\mathbf{X})$ lead to the same distribution, as proven
372 by the following derivation: let $\varphi$ be a bounded and measurable function on $\Theta$.

Using the definitions of the different distributions, we have that:

$$\int_{\Theta} \varphi(\theta) \tilde{\pi}_\lambda(\theta \,|\, \mathbf{X}) d\theta = \int_{\mathbb{R}^p} \varphi(g(\lambda, \varepsilon)) \pi_{\varepsilon, \lambda}(\varepsilon \,|\, \mathbf{X}) d\varepsilon$$

$$= \int_{\mathbb{R}^p} \varphi(g(\lambda, \varepsilon)) \frac{p_\varepsilon(\varepsilon) L_N(X \,|\, g(\lambda, \varepsilon))}{p_\lambda(\mathbf{X})} d\varepsilon$$

$$= \int_{\Theta} \varphi(\theta) \pi_\lambda(\theta) \frac{L_N(\mathbf{X} \,|\, \theta)}{p_\lambda(\mathbf{X})} d\theta$$

$$= \int_{\Theta} \varphi(\theta) \pi_\lambda(\theta \,|\, \mathbf{X}) d\theta.$$

As mentioned in the last Section, when the Jeffreys prior is improper, we compare the posterior distributions, namely, the exact reference posterior when available and the posterior obtained from the VA-RP using the previous method. Altogether, we are able to sample $\theta$ from the posterior even if the density of the parametric prior $\pi_\lambda$ on $\theta$ is unavailable due to an implicit definition of the prior distribution.

For our computations, we choose MCMC sampling, namely an adaptive Metropolis-Hastings sampler with a multivariate Gaussian as the proposition distribution. The adaptation scheme is the following: for each batch of iterations, we monitor the acceptance rate and we adapt the variance parameter of the Gaussian proposition in order to have an acceptance rate close to 40%, which is the advised value (Gelman et al. (2013)) for models in small dimensions. We refer to this algorithm as MH($\varepsilon$). Because we apply MCMC sampling on variable $\varepsilon \in \mathbb{R}^p$ with a reasonable value for $p$, we expect this step of the algorithm to be fast compared to the computation of the VA-RP.

One could also use classic variational inference on $\varepsilon$ instead, but the parametric set of distributions must be chosen wisely. In VAEs for instance, multivariate Gaussian are often considered since it simplifies the KL-divergence term in the ELBO. However, this might be too simplistic in our case since we must apply the neural network $g$ to recover $\theta$ samples. This means that the approximated posterior on $\theta$ belongs to a very similar set of distributions to those used for the VA-RP, since we already used a multivariate Gaussian for the prior on $\varepsilon$. On the other hand, applying once again the implicit sampling approach does not exploit the additional information we have on $\pi_{\varepsilon, \lambda}(\varepsilon \,|\, \mathbf{X})$ compared to $\pi_\lambda(\theta)$, specifically, that its density function is known up to a multiplicative constant. Hence, we argue that using a Metropolis-Hastings sampler is more straightforward in this situation.

## 4 Numerical experiments

We want to apply our algorithm to different statistical models, the first one is the multinomial model, which is the simplest in the sense that the target distributions —the Jeffreys prior and posterior— have explicit expressions and are part of a usual parametric family of proper distributions. The second model —the probit model— will be highlighted with supplementary computations, in regards to the assessment of the stability of our stochastic algorithm, and also with the addition of a moment constraint.

The one-dimensional statistical model of the Gaussian distribution with variance parameter is also presented in Section 6. We stress that this case is a toy model, where the target distributions, namely, the Jeffreys prior and posterior, with or without constraints, can be derived exactly. Essentially, this lets us verify that the output of the algorithm is relevant when compared to the true solution.

Since we only have to compute quotients of the likelihood or the gradient of the log-likelihood, we can omit the multiplicative constant which does not depend on $\theta$.

As for the output of the neural networks, the activation function just before the output is different for each statistical model, the same can be said for the learning rate. In some cases, we apply an affine transformation on the variable $\theta$ to avoid divisions by zero during training. In every test case, we consider simple networks for an easier fine-tuning of the hyperparameters and also because the precise computation of the loss function is an important bottleneck.

For the initialization of the neural networks, biases are set to zero and weights are randomly sampled from a Gaussian distribution. As for the several hyperparameters, we take $N = 10$, $T = 50$ and $U = 1000$ unless stated otherwise. We take a latent space of dimension $p = 50$. For the posterior calculations, we keep the last $5 \cdot 10^4$ samples from the Markov chain over a total of $10^5$ Metropolis-Hastings iterations. Increasing $N$ is advised in order to get closer to the asymptotic case for the optimization problem, and increasing $U$ and $T$ is relevant for the precision of the Monte Carlo estimates. Nevertheless, this increases computation times and we have to do a trade-off between the former and the latter. As for the constrained optimization, we use $\nu = 2$, $M = 0.005$ and $\tilde{\eta}_{max} = 10^4$.

## 4.1 Multinomial model

The multinomial distribution can be interpreted as the generalization of the binomial distribution for higher dimensions. We denote: $X_i \sim \text{Multinomial}(n, (\theta_1, ..., \theta_q))$ with $n \in \mathbb{N}^*$, $\mathbf{X} \in \mathcal{X}^N$ and $\theta \in \Theta$, with: $\mathcal{X} = \{X \in \{0, ..., n\}^q \mid \sum_{j=1}^{q} X^j = n\}$ and $\Theta = \{\theta \in (0, 1)^q \mid \sum_{j=1}^{q} \theta_j = 1\}$. We use $n = 10$ and $q = \dim(\theta) = 4$.

The likelihood function and the gradient of its logarithm are:

$$L_N(\mathbf{X} \mid \theta) = \prod_{i=1}^{N} \frac{n!}{X_i^1! \cdot ... \cdot X_i^q!} \prod_{j=1}^{q} \theta_j^{X_i^j} \propto \prod_{i=1}^{N} \prod_{j=1}^{q} \theta_j^{X_i^j}$$

$$\forall (i, j), \quad \frac{\partial \log L}{\partial \theta_j}(X_i \mid \theta) = \frac{X_i^j}{\theta_j}.$$

The MLE is available: $\forall j$, $\hat{\theta}_{MLE}(j) = \frac{1}{nN} \sum_{i=1}^{N} X_i^j$ and the Jeffreys prior is the $\text{Dir}_q\left(\frac{1}{2}, ..., \frac{1}{2}\right)$ distribution, which is proper. The Jeffreys posterior is a conjugate Dirichlet distribution:

$$J_{post}(\theta \mid \mathbf{X}) = \text{Dir}_q(\theta; \gamma) \quad \text{with} \quad \gamma_j = \frac{1}{2} + \sum_{i=1}^{N} X_i^j.$$

We recall that the probability density function of a Dirichlet distribution of parameter $\gamma$ is the following:

$$\text{Dir}_q(x; \gamma) = \frac{\Gamma(\sum_{j=1}^{q} \gamma_j)}{\prod_{j=1}^{q} \Gamma(\gamma_j)} \prod_{j=1}^{q} x_j^{\gamma_j - 1}.$$

We also use the fact that the marginal distributions of the Dirichlet distribution are Beta distributions, i.e., if $x \sim \text{Dir}_q(\gamma)$, then, for every $j \in \{1, ..., q\}$, $x_j \sim \text{Beta}(\gamma_j, \sum_{k \neq j} \gamma_k)$. The Beta distribution can be seen as a particular case of Dirichlet distribution of dimension $q = 2$.

Although the Jeffreys prior is the prior that maximizes the mutual information, Berger and Bernardo (1992a) and Berger, Bernardo, and Sun (2015) argue that other priors for the multinomial model are more suited in terms of non-informativeness as the dimension of $\theta$ increases. According to them, an appropriate prior is the $m$-group reference prior, where the parameters are grouped into $m$ groups on which a specific ordering is imposed ($1 \leq m \leq q$). The Jeffreys prior is the 1-group reference prior with this definition, while the authors suggest that the $q$-group one is more appropriate. Nevertheless,

our approach consists in approximating the prior yielding the highest mutual information when no ordering is imposed on the parameters, hence, the Jeffreys prior is still the target prior in this regard.

We opt for a simple neural network with one linear layer and a Softmax activation function assuring that all components are positive and sum to 1. Explicitly, we have that:

$$\theta = \text{Softmax}(W\varepsilon + b),$$

with $W \in \mathbb{R}^{4 \times p}$ the weight matrix and $b \in \mathbb{R}^4$ the bias vector. The density function of $\theta$ does not have a closed expression. The following results are obtained with $\alpha = 0.5$ for the divergence and the lower bound is used as the objective function.

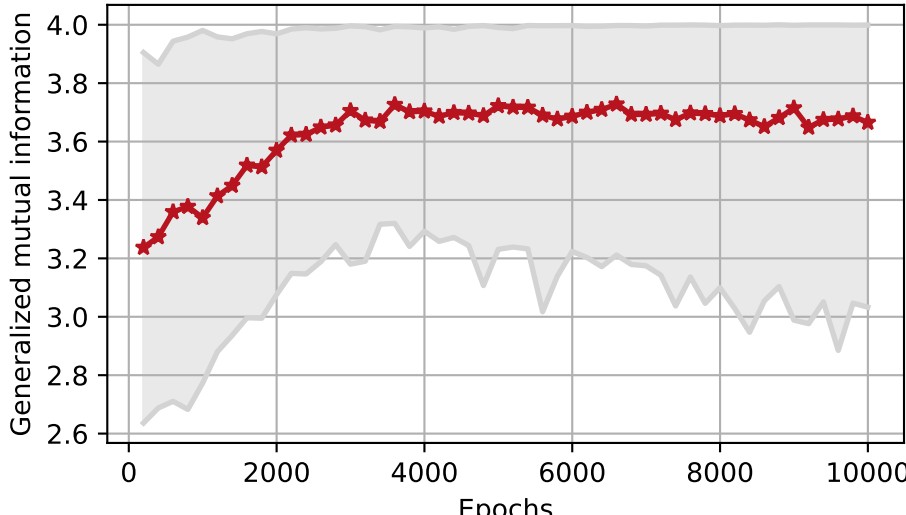

Figure 1: Monte Carlo estimation of the generalized mutual information with $\alpha = 0.5$ (from 200 samples) for $\pi_{\lambda_e}$ where $\lambda_e$ is the parameter of the neural network at epoch $e$. The red curve is the mean value and the gray zone is the 95% confidence interval. The learning rate used in the optimization is 0.0025.

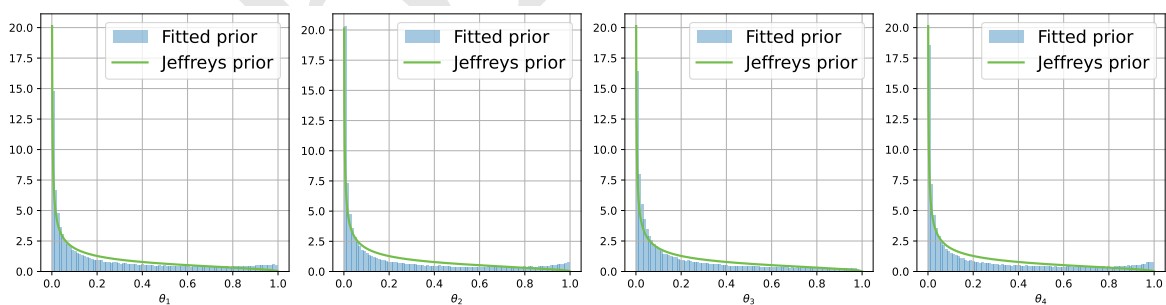

Figure 2: Histograms of the fitted prior and the marginal density functions of the Jeffreys prior $\text{Dir}(\frac{1}{2}, \frac{1}{2}, \frac{1}{2}, \frac{1}{2})$ for each dimension of $\theta$, each histogram is obtained from $10^5$ samples.

For the posterior distribution, we sample 10 times from the Multinomial distribution using $\theta_{true} = (\frac{1}{4}, \frac{1}{4}, \frac{1}{4}, \frac{1}{4})$. The covariance matrix in the proposition distribution of the Metropolis-Hastings algorithm is not diagonal, since we have a relation between the different components of $\theta$, we introduce non-zero covariances. We also verified that the auto-correlation between the successive remaining samples of the Markov chain decreases rapidly on each component.

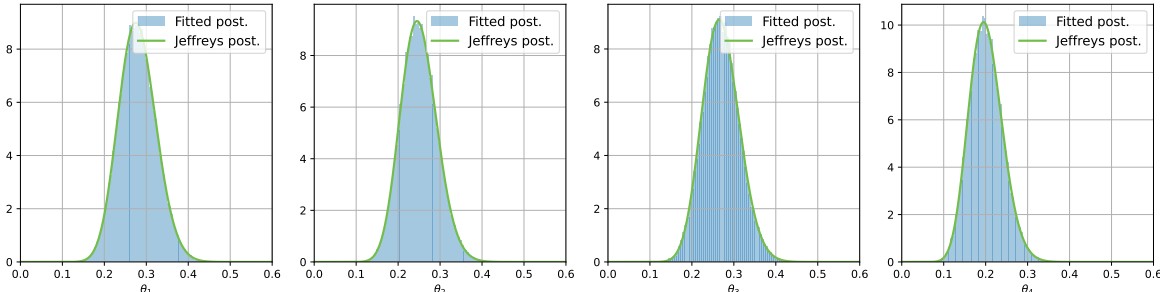

Figure 3: Histograms of the fitted posterior and the marginal density functions of the Jeffreys posterior for each dimension of $\theta$, each histogram is obtained from $5 \cdot 10^4$ samples.

We notice (Figure 1) that the mutual information lies between 0 and $1/\alpha(1-\alpha) = 4$, which is coherent with the theory, the confidence interval is rather large, but the mean value has an increasing trend. In order to obtain more reliable values for the mutual information, one can use more samples in the Monte Carlo estimates at the cost of heavier computations.

Although the shape of the fitted prior resembles the one of the Jeffreys prior, one can notice that it tends to put more weight towards the extremities of the interval (Figure 2). The posterior distribution however is quite similar to the target Jeffreys posterior on every component (Figure 3).

Since the multinomial model is simple and computationally practical, we would like to quantify the effect on the output of the algorithm of some hyperparameters, namely, the divergence parameter $\alpha$, the dimension of the latent space $p$ and the addition of a hidden layer in the neural network. In order to do so, we utilize the maximum mean discrepancy (MMD) defined as:

$$\mathrm{MMD}(\mathbb{P}, \mathbb{Q}) = \|\mu_\mathbb{P} - \mu_\mathbb{Q}\|_\mathscr{H},$$

where $\mu_\mathbb{P}$ and $\mu_\mathbb{Q}$ are respectively the kernel mean embeddings of distributions $\mathbb{P}$ and $\mathbb{Q}$ in a reproducible kernel Hilbert space (RKHS) $(\mathscr{H}, \|\cdot\|_\mathscr{H})$, meaning: $\mu_\mathbb{P}(\theta') = \mathbb{E}_{\theta \sim \mathbb{P}}[K(\theta, \theta')]$ for all $\theta' \in \Theta$ and $K$ being the kernel. The MMD is used for instance in the context of two-sample tests (Gretton et al. (2012)), whose purpose is to compare distributions. We use in our computations the Gaussian or RBF kernel:

$$K(\theta, \theta') = \exp(-0.5 \cdot \|\theta - \theta'\|_2^2),$$

for which the MMD is a metric, this means that the following implication:

$$\mathrm{MMD}(\mathbb{P}, \mathbb{Q}) = 0 \implies \mathbb{P} = \mathbb{Q}$$

is verified with the other axioms. In practice, we consider an unbiased estimator of the $\mathrm{MMD}^2$ given by:

$$\widehat{\mathrm{MMD}^2}(\mathbb{P}, \mathbb{Q}) = \frac{1}{m(m-1)} \sum_{i \neq j} K(x_i, x_j) + \frac{1}{n(n-1)} \sum_{i \neq j} K(y_i, y_j) - \frac{2}{mn} \sum_{i,j} K(x_i, y_j),$$

where $(x_1, ..., x_m)$ and $(y_1, ..., y_n)$ are samples from $\mathbb{P}$ and $\mathbb{Q}$ respectively. In our case, $\mathbb{P}$ is the distribution obtained through variational inference and $\mathbb{Q}$ is the target Jeffreys distribution. Since the MMD can be time-consuming or memory inefficient to compute in practice for very large samples, we consider only the last $2 \cdot 10^4$ entries of our priors and posterior samples.

| $\alpha$ | Prior | Posterior |
|---|---|---|
| 0.10 | $7.07 \times 10^{-2}$ | $2.09 \times 10^{-3}$ |
| 0.25 | $7.42 \times 10^{-2}$ | $3.39 \times 10^{-3}$ |
| 0.50 | $5.26 \times 10^{-2}$ | $1.96 \times 10^{-3}$ |
| 0.75 | $7.80 \times 10^{-2}$ | $1.50 \times 10^{-3}$ |
| 0.90 | $6.15 \times 10^{-2}$ | $4.84 \times 10^{-4}$ |

Table 1: MMD values for different $\alpha$-divergences at prior and posterior levels. As a reference on the prior level, when computing the criterion between two independent Dirichlet $\mathrm{Dir}(\frac{1}{2}, \frac{1}{2}, \frac{1}{2}, \frac{1}{2})$ distributions (i.e. the Jeffreys prior) on $2 \cdot 10^4$ samples, we obtain an order of magnitude of $10^{-3}$. For the posterior level, for which the marginal densities do not diverge at zero, this reference has an order of magnitude of $10^{-4}$.

Firstly, we are interested in the effect of changing the value of $\alpha$ in the $\alpha$-divergence, while keeping $p = 50$ and the same neural network architecture. According to Table 1, the difference between $\alpha$ values in terms of the MMD criterion is essentially inconsequential. One remark is that the mutual information tends to be more unstable as $\alpha$ gets closer to 1. The explanation is that when $\alpha$ tends to 1, we have the approximation:

$$\hat{f}_\alpha(x) \approx \frac{x-1}{\alpha(\alpha-1)} + \frac{x \log(x)}{\alpha},$$

which diverges for all $x$ because of the first term. Hence, we advise the user to avoid $\alpha$ values that are too close to 1. In the following, we use $\alpha = 0.5$ for the divergence.

Secondly, we look at the effect on the dimension of the latent space denoted $p$ for the previously defined neural network architecture, but also when a second layer is added:

$$\theta = \mathrm{Softmax}\left(W_2 \cdot \mathrm{PReLU}_\zeta(W_1 \varepsilon + b_1) + b_2\right),$$

with $W_1 \in \mathbb{R}^{10 \times p}$, $W_2 \in \mathbb{R}^{4 \times 10}$ the weight matrices and $b_1 \in \mathbb{R}^{10}$, $b_2 \in \mathbb{R}^4$ the bias vectors. The added hidden layer is of dimension 10, the activation function between the two layers is the parametric rectified linear unit (PReLU) which is defined as:

$$\mathrm{PReLU}_\zeta(x) = \begin{cases} x \text{ if } x \geq 0 \\ \zeta x \text{ if } x < 0, \end{cases}$$

with $\zeta > 0$ a learnable parameter. The activation function is applied element-wise.

| $p$ | Prior (1 layer) | Posterior (1 layer) | Prior (2 layers) | Posterior (2 layers) |
|---|---|---|---|---|
| 25 | $8.16 \times 10^{-2}$ | $2.02 \times 10^{-3}$ | $2.43 \times 10^{-1}$ | $2.80 \times 10^{-2}$ |
| 50 | $5.26 \times 10^{-2}$ | $1.96 \times 10^{-3}$ | $3.23 \times 10^{-1}$ | $7.09 \times 10^{-2}$ |
| 75 | $5.35 \times 10^{-2}$ | $3.79 \times 10^{-3}$ | $2.59 \times 10^{-1}$ | $1.41 \times 10^{-2}$ |
| 100 | $3.21 \times 10^{-2}$ | $2.75 \times 10^{-3}$ | $2.41 \times 10^{-1}$ | $1.47 \times 10^{-2}$ |
| 200 | $4.02 \times 10^{-2}$ | $1.84 \times 10^{-3}$ | $2.10 \times 10^{-1}$ | $2.71 \times 10^{-2}$ |

| $p$ | Prior (1 layer) | Posterior (1 layer) | Prior (2 layers) | Posterior (2 layers) |
|---|---|---|---|---|

Table 2: MMD values for different $\alpha$-divergences at prior and posterior levels. As a reference on the prior level, when computing the criterion between two independent Dirichlet $\mathrm{Dir}(\frac{1}{2}, \frac{1}{2}, \frac{1}{2}, \frac{1}{2})$ distributions (i.e. the Jeffreys prior) on $2 \cdot 10^4$ samples, we obtain an order of magnitude of $10^{-3}$. For the posterior level, for which the marginal densities do not diverge at zero, this reference has an order of magnitude of $10^{-4}$.

Several observations can be made thanks to Table 2. Firstly, looking at the table column-wise, one can notice that the value of $p$ tends to have little influence on the MMD values, since the order of magnitude always remains the same for each column. We remark also that the MMD values for the simple neural network with one layer are always lower than those for the neural network with the additional hidden layer when reading the table row-wise. This is true for all values of $p$ at both the prior and the posterior level. It is important to note that these experiments were conducted with fixed values of $T$ and $U$, which determine the number of samples used in the Monte Carlo approximation of the objective function's gradient. We note that increasing $T$ and $U$ could improve the quality of VA-RP approximations for more complex networks. However, doing so exponentially increases the computational cost of the method.

## 4.2 Probit model

We present in this section the probit model used to estimate seismic fragility curves, which was introduced by Kennedy et al. (1980), it is also referred as the log-normal model in the literature. A seismic fragility curve is the probability of failure $P_f(a)$ of a mechanical structure subjected to a seism as a function of a scalar value $a$ derived from the seismic ground motion. The properties of the Jeffreys prior for this model are highlighted by Van Biesbroeck et al. (2024).

The model is defined by the observation of an i.i.d. sample $\mathbf{X} = (X_1, \dots, X_N)$ where for any $i$, $X_i \sim (Z, a) \in \mathcal{X} = \{0, 1\} \times (0, \infty)$. The distribution of the r.v. $(Z, a)$ is parameterized by $\theta = (\theta_1, \theta_2) \in (0, \infty)^2$ as:

$$\begin{cases} a \sim \mathrm{Log\text{-}}\mathcal{N}(\mu_a, \sigma_a^2) \\ P_f(a) = \Phi\left(\dfrac{\log a - \log \theta_1}{\theta_2}\right) \\ Z \sim \mathrm{Bernoulli}(P_f(a)), \end{cases}$$

where $\Phi$ is the cumulative distribution function of the standard Gaussian. The probit function is the inverse of $\Phi$. The likelihood is of the form:

$$\begin{cases} L_N(\mathbf{X} \mid \theta) = \displaystyle\prod_{i=1}^{N} p(a_i) \prod_{i=1}^{N} P_f(a_i)^{Z_i}(1 - P_f(a_i))^{1-Z_i} \propto \prod_{i=1}^{N} P_f(a_i)^{Z_i}(1 - P_f(a_i))^{1-Z_i} \\ p(a_i) = \dfrac{1}{a_i\sqrt{2\pi\sigma_a^2}} \exp\left(-\dfrac{1}{2\sigma_a^2}(\log a_i - \mu_a)^2\right). \end{cases}$$

For simplicity, we denote: $\gamma_i = \dfrac{\log a_i - \log \theta_1}{\theta_2} = \Phi^{-1}(P_f(a_i)) = \mathrm{probit}(P_f(a_i))$, the gradient of the log-likelihood is the following:

$$\begin{cases} \dfrac{\partial \log L_N}{\partial \theta_1}(\mathbf{X}\,|\,\theta) = \displaystyle\sum_{i=1}^{N} \dfrac{1}{\theta_1 \theta_2} \left( (-Z_i)\dfrac{\Phi'(\gamma_i)}{\Phi(\gamma_i)} + (1-Z_i)\dfrac{\Phi'(\gamma_i)}{1-\Phi(\gamma_i)} \right) \\ \dfrac{\partial \log L_N}{\partial \theta_2}(\mathbf{X}\,|\,\theta) = \displaystyle\sum_{i=1}^{N} \dfrac{\gamma_i}{\theta_2} \left( (-Z_i)\dfrac{\Phi'(\gamma_i)}{\Phi(\gamma_i)} + (1-Z_i)\dfrac{\Phi'(\gamma_i)}{1-\Phi(\gamma_i)} \right). \end{cases}$$

There is no explicit formula for the MLE, so it has to be approximated using samples. This statistical model is a more difficult case than the previous one, since no explicit formula for the Jeffreys prior is available either but it has been shown by Van Biesbroeck et al. (2024) that it is improper in $\theta_2$ and some asymptotic rates where derived. More precisely, when $\theta_1 > 0$ is fixed,

$$\begin{cases} J(\theta) \propto 1/\theta_2 & \text{as} \quad \theta_2 \longrightarrow 0 \\ J(\theta) \propto 1/\theta_2^3 & \text{as} \quad \theta_2 \longrightarrow +\infty. \end{cases}$$

If we fix $\theta_2 > 0$, the prior is proper for the variable $\theta_1$:

$$J(\theta) \propto \dfrac{|\log \theta_1|}{\theta_1} \exp\left( -\dfrac{(\log \theta_1 - \mu_a)^2}{2\theta_2 + 2\sigma_a^2} \right) \quad \text{when} \quad |\log \theta_1| \longrightarrow +\infty.$$

which resembles a log-normal distribution except for the $|\log \theta_1|$ factor. Since the density of the Jeffreys prior is not explicit and can not be computed directly, the Fisher information matrix is computed in Van Biesbroeck et al. (2024) using numerical integration with Simpson's rule on a specific grid and then an interpolation is applied. We use this computation as the reference to evaluate the quality of the output of our algorithm. In the mentioned article, the posterior distribution is also computed with an adaptive Metropolis-Hastings algorithm on the variable $\theta$, we refer to this algorithm as MH($\theta$) since it is different from the one mentioned in Section 3.4. More details on MH($\theta$) are given in Gauchy (2022). We take $\mu_a = 0$, $\sigma_a^2 = 1$, $N = 500$ and $U = 500$ for the computation of the prior.

As for the neural network, we use a one-layer network with an exp activation for $\theta_1$ and a Softplus activation for $\theta_2$. We have that:

$$\theta = \begin{pmatrix} \theta_1 \\ \theta_2 \end{pmatrix} = \begin{pmatrix} \exp(w_1^\top \varepsilon + b_1) \\ \log\left(1 + \exp(w_2^\top \varepsilon + b_2)\right) \end{pmatrix},$$

with $w_1, w_2 \in \mathbb{R}^p$ the weight vectors and $b_1, b_2 \in \mathbb{R}$ the biases, thus we have $\lambda = (w_1, w_2, b_1, b_2)$. Because this architecture remains simple, it is possible to elucidate the resulting marginal distributions of $\theta_1$ and $\theta_2$. The first component $\theta_1$ follows a Log-$\mathcal{N}(b_1, \|w_1\|_2^2)$ distribution and $\theta_2$ has an explicit density function:

$$p(\theta_2) = \dfrac{1}{\sqrt{2\pi \|w_2\|_2^2}(1 - e^{-\theta_2})} \exp\left( -\dfrac{1}{2\|w_2\|_2^2} \left( \log(e^{\theta_2} - 1) - b_2 \right)^2 \right).$$

These expressions describe the parameterized set $\mathscr{P}_\Lambda$ of priors considered in the optimization problem. This set is restrictive, so that the resulting VA-RP must be interpreted as the most objective —according to the mutual information criterion— prior among the ones in $\mathscr{P}_\Lambda$. Since we do not know any explicit expression of the Jeffreys prior for this prior, we cannot provide a precise comparison between the parameterized VA-RP elucidated above and the target. However, the form of the distribution of

$\theta_1$ qualitatively resembles its theoretical target. In the case of $\theta_2$, the asymptotic decay rates of its density function can be derived:

$$\begin{cases} p(\theta_2) \underset{\theta_2 \to 0}{=} \frac{1}{\theta_2 \sqrt{2\pi}\|w_2\|_2} \exp\left(-\frac{(\log \theta_2 - b_2)^2}{2\|w_2\|_2^2}\right); \\ p(\theta_2) \underset{\theta_2 \to \infty}{=} \frac{1}{\sqrt{2\pi}\|w_2\|_2} \exp\left(-\frac{(\theta_2 - b_2)^2}{2\|w_2\|_2^2}\right). \end{cases} \tag{12}$$

While $\|w_2\|_2$ does not tend toward $\infty$, these decay rates strongly differ from the ones of the Jeffreys prior w.r.t. $\theta_2$. Otherwise, the decay rates resemble to something proportional to $(\theta_2 + 1)^{-1}$ in both directions. In our numerical computations, the optimization process yielded a VA-RP with parameters $w_2$ and $b_2$ that did not diverge to extreme values.

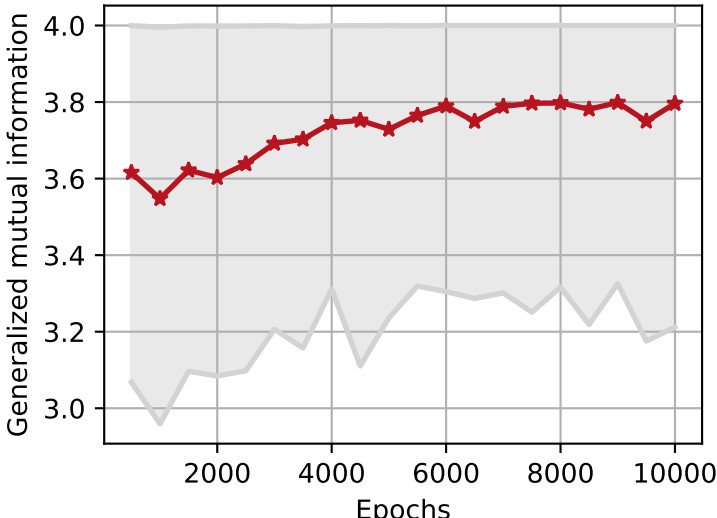

Figure 4: Monte Carlo estimation of the generalized mutual information with $\alpha = 0.5$ (from 100 samples) for $\pi_{\lambda_e}$ where $\lambda_e$ is the parameter of the neural network at epoch $e$. The red curve is the mean value and the gray zone is the 95% confidence interval. The learning rate used in the optimization is 0.001.

In Figure 4 is shown the evolution of the mutual information through the optimization of the VA-RP for the probit model. We perceive high mutual information values at the initialization, which we interpret as a result of the fact that the parametric prior on $\theta_1$ is already quite close to its target distribution.

With $\alpha$-divergences, using a moment constraint of the form $a(\theta_2) = \theta_2^\kappa$ for the second component is relevant here as long as $\kappa \in \left(0, \frac{2}{1+1/\alpha}\right)$, to ensure that the resulting constrained prior is indeed proper. With $\alpha = 0.5$, we take the value $\kappa = 1/8$ and we use the same neural network. The evolution of the mutual information through the optimization of the constrained VA-RP is proposed in Figure 5. In Figure 6 is presented the evolution of the constrained gap: the difference between the target and current values for the constraint.

For the posterior, we take as dataset 50 samples from the probit model with $\theta_{true}$ close to $(3.37, 0.43)$. For computational reasons, the Metropolis-Hastings algorithm is applied for only $5 \cdot 10^4$ iterations. An important remark is that if the size of the dataset is rather small, the probability that the data is degenerate is not negligible. By degenerate data, we refer to situations when the data points are partitioned into two disjoint subsets when classified according to $a$ values, the posterior becomes improper because the likelihood is constant (Van Biesbroeck et al. (2024)). In such cases, the

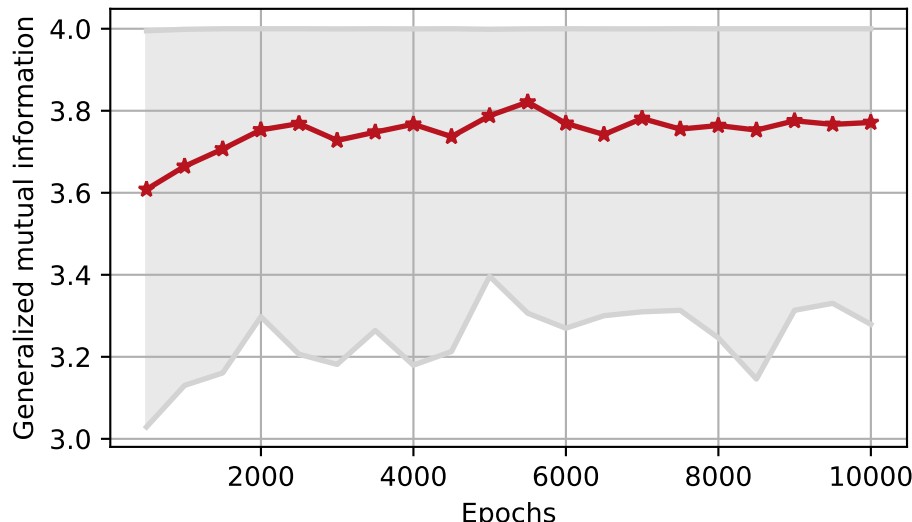

Figure 5: Monte Carlo estimation of the generalized mutual information with $\alpha = 0.5$ (from 100 samples) for $\pi_{\lambda_e}$ where $\lambda_e$ is the parameter of the neural network at epoch $e$. The red curve is the mean value and the gray zone is the 95% confidence interval. The learning rate used in the optimization is 0.0005.

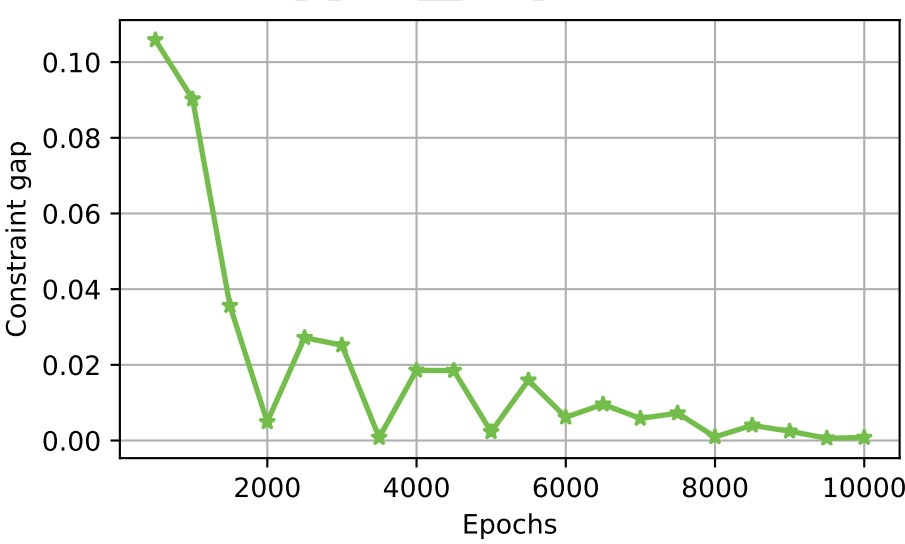

Figure 6: Evolution of the constraint value gap during training. It corresponds to the difference between the target and current values for the constraint (in absolute value)

convergence of the Markov chains is less apparent, the plots for this section are obtained with non-degenerate datasets.

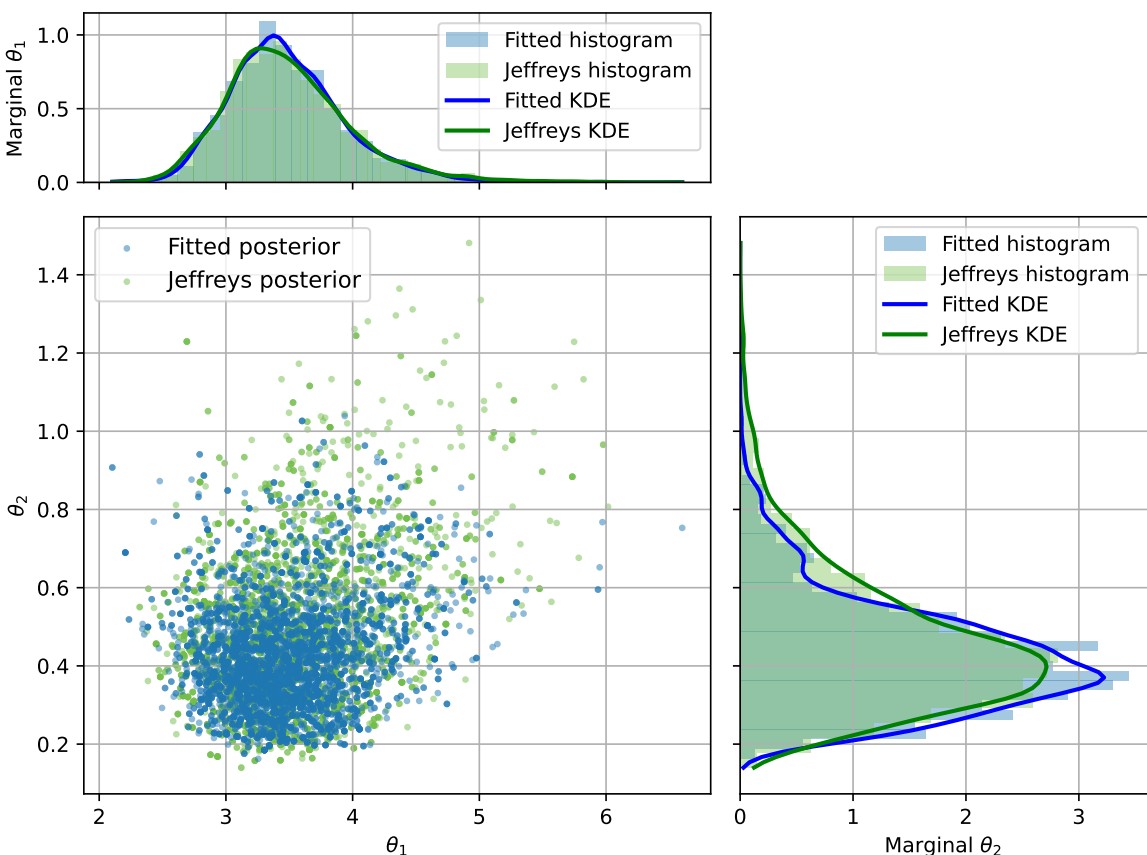

Figure 7: Scatter histogram of the unconstrained fitted posterior and the Jeffreys posterior distributions obtained from 5000 samples. Kernel density estimation is used on the marginal distributions in order to approximate their density functions with Gaussian kernels.

As Figure 7 shows, we obtain a relevant approximation of the true Jeffreys posterior especially on the variable $\theta_1$, whereas a small difference is present for the tail of the distribution on $\theta_2$. The latter remark was expected regarding the analytical study of the marginal distribution of $\pi_\lambda$ w.r.t. $\theta_2$ given the architecture considered for the VA-RP (see Equation 12). It is interesting to see that the difference between the posteriors is harder to discern in the neighborhood of $\theta_2 = 0$. Indeed, in such case where the data are not degenerate, the likelihood provides a strong decay rate when $\theta_2 \to 0$ that makes the influence of the prior negligible (see Van Biesbroeck et al. (2024)):

$$L_N(\mathbf{X}\,|\,\theta) \underset{\theta_2 \to 0}{=} \theta_2^{\|\chi\|_2^2} \exp\left(-\frac{1}{2\theta_2^2}\sum_{i=1}^{N}\chi_i(\log a_i - \log \theta_1)^2\right),$$

where $\chi \in \{0, 1\}^N$ is a non-null vector that depends on $\mathbf{X}$.

When $\theta_2 \to \infty$, however, the likelihood does not reduce the influence of the prior as it remains asymptotically constant: $L_N(\mathbf{X}\,|\,\theta) \underset{\theta_2 \to \infty}{\to} 2^{-N}$.

The result on the constrained case (Figure 8) is very similar to the unconstrained one.

Altogether, one can observe that the variational inference approach yields close results to the numerical integration approach (Van Biesbroeck et al. (2024)), with or without constraints, even

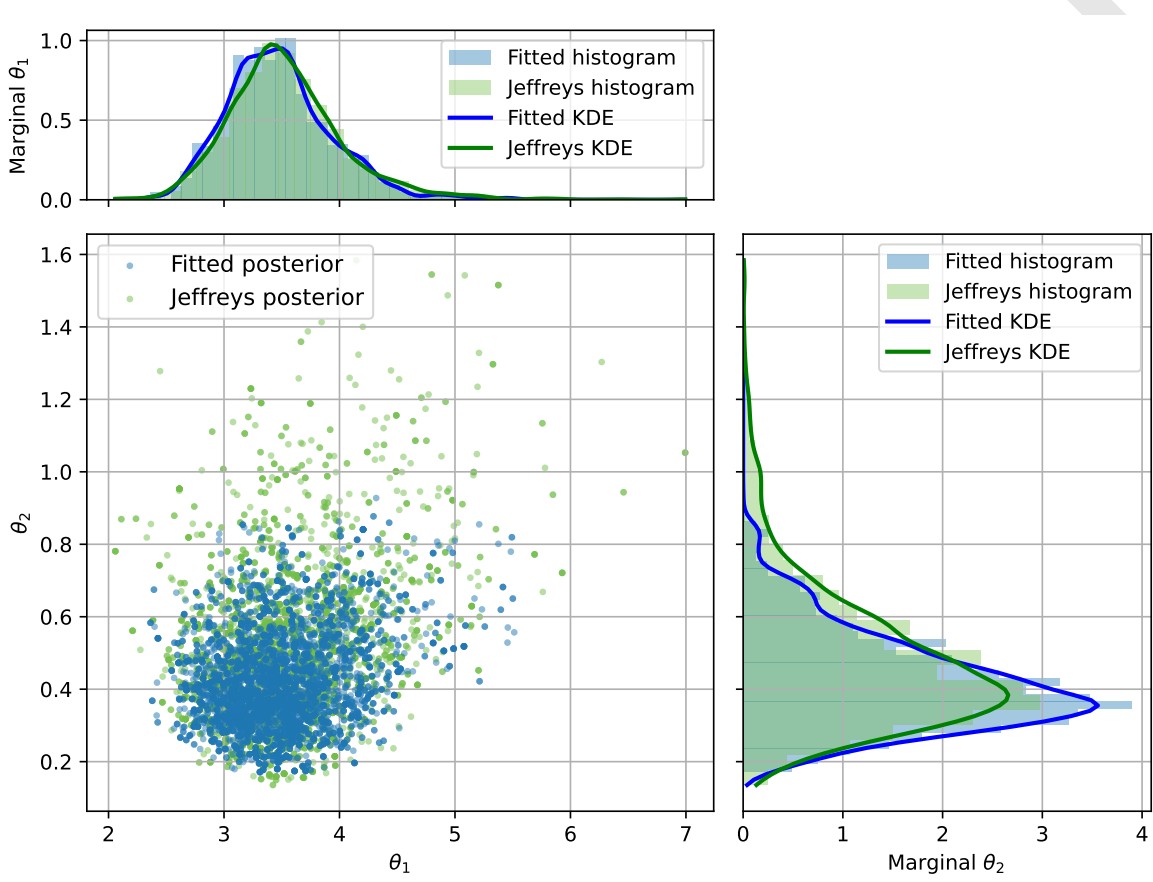

Figure 8: Scatter histogram of the constrained fitted posterior and the target posterior distributions obtained from 5000 samples. Kernel density estimation is used on the marginal distributions in order to approximate their density functions with Gaussian kernels.

though the matching of the decay rates w.r.t. $\theta_2$ remains limited given the simple network that we have used in this case.

To ascertain the relevancy of our posterior approximation, we compute the posterior mean euclidean norm difference $\mathbb{E}_\theta[\|\theta - \theta_{true}\|]$ as a function of the size of the dataset. In each computation, the neural network remains the same but the dataset changes by adding new entries.

Furthermore, in order to assess the stability of the stochastic optimization with respect to the random number generator (RNG) seed, we also compute the empirical cumulative distribution functions (ECDFs) for each posterior distribution. For every seed, the parameters of the neural network are expected to be different, we keep the same dataset for the MCMC sampling however.

Finally, we compute the ECDFs for different values of the dimension of the latent space $p$ in order to quantify the sensitivity of the output distributions with respect to this hyperparameter.

These computations are done in the unconstrained case as well as the constrained one. The different plots and details can be found in Section 6.

# 5   Conclusion

In this work, we developed an algorithm to perform variational approximation of objective priors using a generalized definition of mutual information based on $f$-divergences. To enhance computational efficiency, we derived a lower bound of the generalized mutual information. Additionally, because the objective priors of interest, which are Jeffreys priors, often yield improper posteriors, we adapted the variational definition of the problem to incorporate constraints that ensure the posteriors are proper.

Numerical experiments have been carried out on various test cases of different complexities in order to validate our approach. These test cases range from purely toy models to more real-world problems, namely the estimation of seismic fragility curve parameters using a probit statistical model. The results demonstrate the usefulness of our approach in estimating both prior and posterior distributions across various problems, including problems where the theoretical expression of the target prior is cumbersome to compute.

Our development is supported by an open source and flexible implementation, which can be adapted to a wide range of statistical models.

Looking forward, the approximation of the tails of the reference priors should be improved, but this is a complex and general problem in the field of variational approximation. Furthermore, the stability of the algorithm which seems to depend on the statistical model and the architecture of the neural network is an other issue to be addressed. An extension of this work to the approximation of Maximal Data Information (MDI) priors is also appealing, thanks to the fact that MDI are proper under certain assumptions precised in Bousquet (2008).

# Acknowledgement

This research was supported by the CEA (French Alternative Energies and Atomic Energy Commission) and the SEISM Institute (https://www.institut-seism.fr/en/).

## 6 Appendix

### 6.1 Gradient computation of the generalized mutual information

We recall that $F(x) = f(x) - xf'(x)$ and $p_\lambda$ is a shortcut notation for $p_{\pi_\lambda,N}$ being the marginal distribution under $\pi_\lambda$. The generalized mutual information writes:

$$I_{D_f}(\pi_\lambda; L_N) = \int_\Theta D_f(p_\lambda \| L_N(\cdot \,|\, \theta))\pi_\lambda(\theta)d\theta$$

$$= \int_\Theta \int_{\mathcal{X}^N} \pi_\lambda(\theta)L_N(\mathbf{X}\,|\,\theta)f\left(\frac{p_\lambda(\mathbf{X})}{L_N(\mathbf{X}\,|\,\theta)}\right)d\mathbf{X}d\theta.$$

For each $l$, taking the derivative with respect to $\lambda_l$ yields:

$$\frac{\partial I_{D_f}}{\partial \lambda_l}(\pi_\lambda; L_N) = \int_\Theta \int_{\mathcal{X}^N} \frac{\partial \pi_\lambda}{\partial \lambda_l}(\theta)L_N(\mathbf{X}\,|\,\theta)f\left(\frac{p_\lambda(\mathbf{X})}{L_N(\mathbf{X}\,|\,\theta)}\right)d\mathbf{X}d\theta$$

$$+ \int_\Theta \int_{\mathcal{X}^N} \pi_\lambda(\theta)L_N(\mathbf{X}\,|\,\theta)\frac{\partial p_\lambda}{\partial \lambda_l}\frac{1}{L_N(\mathbf{X}\,|\,\theta)}(\mathbf{X})f'\left(\frac{p_\lambda(\mathbf{X})}{L_N(\mathbf{X}\,|\,\theta)}\right)d\mathbf{X}d\theta,$$

or in terms of expectations:

$$\frac{\partial I_{D_f}}{\partial \lambda_l}(\pi_\lambda; L_N) = \frac{\partial}{\partial \lambda_l}\mathbb{E}_{\theta \sim \pi_\lambda}\left[\tilde{I}(\theta)\right] + \mathbb{E}_{\theta \sim \pi_\lambda}\left[\mathbb{E}_{\mathbf{X} \sim L_N(\cdot|\theta)}\left[\frac{1}{L_N(\mathbf{X}\,|\,\theta)}\frac{\partial p_\lambda}{\partial \lambda_l}(\mathbf{X})f'\left(\frac{p_\lambda(\mathbf{X})}{L_N(\mathbf{X}\,|\,\theta)}\right)\right]\right],$$

where:

$$\tilde{I}(\theta) = \int_{\mathcal{X}^N} L_N(\mathbf{X}\,|\,\theta)f\left(\frac{p_\lambda(\mathbf{X})}{L_N(\mathbf{X}\,|\,\theta)}\right)d\mathbf{X}.$$

We note that the derivative with respect to $\lambda_l$ does not apply to $\tilde{I}$ in the previous equation. Using the chain rule yields:

$$\frac{\partial}{\partial \lambda_l}\mathbb{E}_{\theta \sim \pi_\lambda}\left[\tilde{I}(\theta)\right] = \frac{\partial}{\partial \lambda_l}\mathbb{E}_\varepsilon\left[\tilde{I}(g(\lambda, \varepsilon))\right] = \mathbb{E}_\varepsilon\left[\sum_{j=1}^q \frac{\partial \tilde{I}}{\partial \theta_j}(g(\lambda, \varepsilon))\frac{\partial g_j}{\partial \lambda_l}(\lambda, \varepsilon)\right].$$

We have the following for every $j \in \{1, ..., q\}$:

$$\frac{\partial \tilde{I}}{\partial \theta_j}(\theta) = \int_{\mathcal{X}^N} \frac{-p_\lambda(\mathbf{X})}{L_N(\mathbf{X}\,|\,\theta)}\frac{\partial L_N}{\partial \theta_j}(\mathbf{X}\,|\,\theta)f'\left(\frac{p_\lambda(\mathbf{X})}{L_N(\mathbf{X}\,|\,\theta)}\right) + f\left(\frac{p_\lambda(\mathbf{X})}{L_N(\mathbf{X}\,|\,\theta)}\right)\frac{\partial L_N}{\partial \theta_j}(\mathbf{X}\,|\,\theta)d\mathbf{X}$$

$$= \int_{\mathcal{X}^N} F\left(\frac{p_\lambda(\mathbf{X})}{L_N(\mathbf{X}\,|\,\theta)}\right)\frac{\partial L_N}{\partial \theta_j}(\mathbf{X}\,|\,\theta)d\mathbf{X}$$

$$= \mathbb{E}_{\mathbf{X} \sim L_N(\cdot|\theta)}\left[\frac{\partial \log L_N}{\partial \theta_j}(\mathbf{X}\,|\,\theta)F\left(\frac{p_\lambda(\mathbf{X})}{L_N(\mathbf{X}\,|\,\theta)}\right)\right].$$

Putting everything together, we finally obtain the desired formula. The gradient of the generalized lower bound function is obtained in a very similar manner.

In what follows, we prove that the gradient of $I_{D_f}$, as formulated in Equation 10 aligns with the form of Equation 9. We write, for $l \in \{1, ..., L\}$:

$$\frac{\partial I_{D_f}}{\partial \lambda_l}(\pi_\lambda; L_N) = \mathbb{E}_\varepsilon\left[\sum_{j=1}^q \frac{\partial \tilde{I}}{\partial \theta_j}(g(\lambda, \varepsilon))\frac{\partial g_j}{\partial \lambda_l}(\lambda, \varepsilon)\right] + \mathcal{G}_l,$$

where

$$\mathscr{G}_l = \mathbb{E}_{\theta \sim \pi_\lambda} \mathbb{E}_{\mathbf{X} \sim L_N(\cdot|\theta)} \left[ \frac{1}{L_N(\mathbf{X}|\theta)} \frac{\partial p_\lambda}{\partial \lambda_l}(\mathbf{X}) f'\left(\frac{p_\lambda(\mathbf{X})}{L_N(\mathbf{X}|\theta)}\right) \right].$$

We remark that

$$\frac{\partial p_\lambda}{\partial \lambda_l}(\mathbf{X}) = \mathbb{E}_{\varepsilon_2} \sum_{j=1}^q \frac{\partial L_N}{\partial \theta_j}(\mathbf{X}|g(\lambda, \varepsilon_2)) \frac{\partial g_j}{\partial \lambda_l}(\lambda, \varepsilon_2).$$

Thus, we can develop $\mathscr{G}_l$ as:

$$
\begin{aligned}
\mathscr{G}_l =& \mathbb{E}_{\varepsilon_1} \mathbb{E}_{\mathbf{X} \sim L_N(\cdot|g(\lambda, \varepsilon_1))} \mathbb{E}_{\varepsilon_2} \sum_j \frac{1}{L_N(\mathbf{X}|g(\lambda, \varepsilon_1))} f'\left(\frac{p_\lambda(\mathbf{X})}{L_N(\mathbf{X}|g(\lambda, \varepsilon_1))}\right) \frac{\partial L_N}{\partial \theta_j}(\mathbf{X}|g(\lambda, \varepsilon_2)) \frac{\partial g_j}{\partial \lambda_l}(\lambda, \varepsilon_2) \\
=& \mathbb{E}_{\varepsilon_2} \mathbb{E}_{\varepsilon_1} \mathbb{E}_{\mathbf{X} \sim L_N(\cdot|g(\lambda, \varepsilon_1))} \sum_j \frac{1}{L_N(\mathbf{X}|g(\lambda, \varepsilon_1))} f'\left(\frac{p_\lambda(\mathbf{X})}{L_N(\mathbf{X}|g(\lambda, \varepsilon_1))}\right) \frac{\partial L_N}{\partial \theta_j}(\mathbf{X}|g(\lambda, \varepsilon_2)) \frac{\partial g_j}{\partial \lambda_l}(\lambda, \varepsilon_2) \\
=& \mathbb{E}_{\varepsilon_2} \sum_{j=1}^q \frac{\partial g_j}{\partial \lambda_l}(\lambda, \varepsilon_2) \mathbb{E}_{\varepsilon_1} \mathbb{E}_{\mathbf{X} \sim L_N(\cdot|g(\lambda, \varepsilon_1))} \frac{1}{L_N(\mathbf{X}|g(\lambda, \varepsilon_1))} f'\left(\frac{p_\lambda(\mathbf{X})}{L_N(\mathbf{X}|g(\lambda, \varepsilon_1))}\right) \frac{\partial L_N}{\partial \theta_j}(\mathbf{X}|g(\lambda, \varepsilon_2)).
\end{aligned}
$$

Now, calling $\tilde{K}$ the function defined as follows:

$$\tilde{K} \, : \, \theta \mapsto \tilde{K}(\theta) = \mathbb{E}_{\varepsilon_1} \mathbb{E}_{\mathbf{X} \sim L_N(\cdot|g(\lambda, \varepsilon_1))} \left[ \frac{1}{L_N(\mathbf{X}|g(\lambda, \varepsilon_1))} f'\left(\frac{p_\lambda(\mathbf{X})}{L_N(\mathbf{X}|g(\lambda, \varepsilon_1))}\right) L_N(\mathbf{X}|\theta) \right],$$

we obtain that

$$\mathscr{G}_l = \mathbb{E}_{\varepsilon_2} \sum_{j=1}^q \frac{\partial g_j}{\partial \lambda_l}(\lambda, \varepsilon_2) \frac{\partial \tilde{K}}{\partial \theta_j}(g(\lambda, \varepsilon_2)).$$

Eventually, denoting $\tilde{\mathbf{I}} = \tilde{K} + \tilde{I}$, we have:

$$\frac{\partial I_{D_f}}{\partial \lambda_l}(\pi_\lambda; L_N) = \mathbb{E}_\varepsilon \left[ \sum_{j=1}^q \frac{\partial \tilde{\mathbf{I}}}{\partial \theta_j}(g(\lambda, \varepsilon)) \frac{\partial g_j}{\partial \lambda_l}(\lambda, \varepsilon) \right],$$

which is compatible with the form of Equation 9.

## 6.2 Gaussian distribution with variance parameter

We consider a normal distribution where $\theta$ is the variance parameter: $X_i \sim \mathcal{N}(\mu, \theta)$ with $\mu \in \mathbb{R}$, $\mathbf{X} \in \mathcal{X}^N = \mathbb{R}^N$ and $\theta \in \mathbb{R}_+^*$. We take $\mu = 0$. The likelihood and score functions are:

$$L_N(\mathbf{X} \,|\, \theta) = \prod_{i=1}^N \frac{1}{\sqrt{2\pi\theta}} \exp\left(-\frac{1}{2\theta}(X_i - \mu)^2\right)$$

$$\frac{\partial \log L_N}{\partial \theta}(\mathbf{X} \,|\, \theta) = -\frac{N}{2\theta} + \frac{1}{2\theta^2} \sum_{i=1}^N (X_i - \mu)^2.$$

The MLE is available: $\hat{\theta}_{MLE} = \frac{1}{N} \sum_{i=1}^N X_i$. However, the Jeffreys prior is an improper distribution in this case: $J(\theta) \propto 1/\theta$. Nevertheless, the Jeffreys posterior is a proper inverse-gamma distribution:

$$J_{post}(\theta \,|\, \mathbf{X}) = \Gamma^{-1}\left(\theta; \frac{N}{2}, \frac{1}{2} \sum_{i=1}^N (X_i - \mu)^2\right).$$

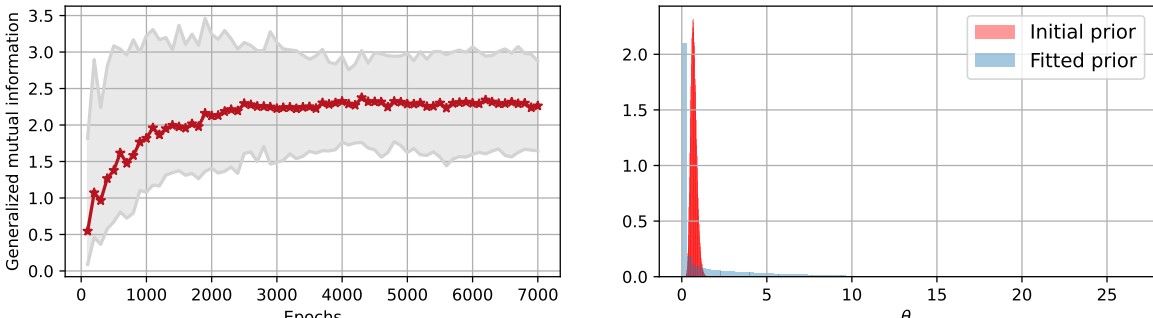

Figure 9: Left: Monte Carlo estimation of the generalized mutual information with $\alpha = 0.5$ (from 200 samples) for $\pi_{\lambda_e}$ where $\lambda_e$ is the parameter of the neural network at epoch $e$. The red curve is the mean value and the gray zone is the 95% confidence interval. Right: Histograms of the initial prior (at epoch 0) and the fitted prior (after training), each one is obtained from $10^5$ samples. The learning rate used in the optimization is 0.025.

We use a neural network with one layer and a Softplus activation function. The dimension of the latent variable $\varepsilon$ is $p = 10$.

We retrieve close results to those of Gauchy et al. (2023), even though we used the $\alpha$-divergence instead of the classic KL-divergence (Figure 9). The evolution of the mutual information seems to be more stable during training. We can not however directly compare our result to the target Jeffrey prior since the latter is improper.

For the posterior distribution, we sample 10 times from the normal distribution using $\theta_{true} = 1$.

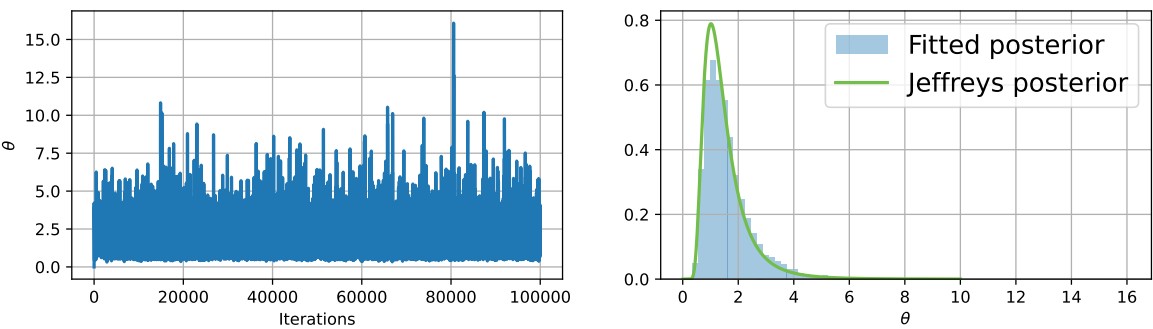

Figure 10: Left: Markov chain during the Metropolis-Hastings iterations. Right: Histogram of the fitted posterior obtained from $5 \cdot 10^4$ samples and the density function of the Jeffreys posterior.

As Figure 10 shows, we obtain a parametric posterior distribution which closely resembles the target distribution, even though the theoretical prior is improper.

In order to evaluate the performance of the algorithm for the prior, we have to add a constraint. The simplest kind of constraints are moment constraints with: $a(\theta) = \theta^\beta$, however, we can not use such a constraint here since the integrals for $\mathscr{K}$ and $c$ from Section 2 would diverge either at 0 or at $+\infty$.

If we define: $a(\theta) = \dfrac{1}{\theta^\beta + \theta^\tau}$ with $\beta < 0 < \tau$, then the integrals for $\mathscr{K}$ and $c$ are finite, because:

$$\forall \delta \geq 1, \quad \int_0^{+\infty} \frac{1}{\theta} \cdot \left( \frac{1}{\theta^\beta + \theta^\tau} \right)^\delta d\theta \leq \frac{1}{\delta} \left( \frac{1}{\tau} - \frac{1}{\beta} \right).$$

This function of constraint $a$ is preferable because it yields different asymptotic rates at 0 and $+\infty$:

$$\begin{cases} a(\theta) \sim \theta^{-\beta} & \text{as} \quad \theta \longrightarrow 0 \\ a(\theta) \sim \theta^{-\tau} & \text{as} \quad \theta \longrightarrow +\infty. \end{cases}$$

In order to apply the algorithm, we are interested in finding:

$$\mathcal{K} = \int_0^{+\infty} \frac{1}{\theta} \cdot a(\theta)^{1/\alpha} d\theta \quad \text{and} \quad c = \int_0^{+\infty} \frac{1}{\theta} \cdot a(\theta)^{1+(1/\alpha)} d\theta.$$

For instance, let $\alpha = 1/2$. If $\beta = -1$, $\tau = 1$, then $\mathcal{K} = 1/2$ and $c = \pi/16$. The constraint value is $c/\mathcal{K} = \pi/8$. Thus, for this example, we only have to apply the third step of the proposed method. We use in this case a one-layer neural network with exp as the activation function, the parametric set of priors corresponds to log-normal distributions.

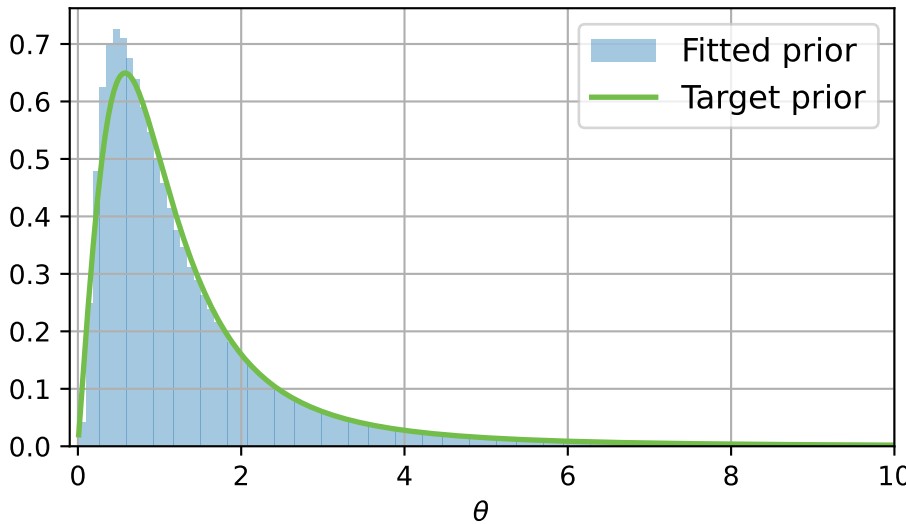

Figure 11: Histogram of the constrained fitted prior obtained from $10^5$ samples, and density function of the target prior. The learning rate used in the optimization is 0.0005.

In this case we are able to compare prior distributions since both are proper, as Figure 11 shows, we recover a relevant result using our algorithm even with added constraints.

The density function of the posterior is known up to a multiplicative constant, more precisely, it corresponds to the product of the constraint function and the density function of an inverse-gamma distribution. Hence, the constant can be estimated using Monte Carlo samples from the inverse-gamma distribution in question. We apply the same approach as before in order to obtain the posterior from the parametric prior.

As shown in Figure 12, the parametric posterior has a shape similar to the theoretical distribution.

### 6.3 Probit model and robustness

As mentioned in Section 4.2 regarding the probit model, we present several additional results.

Figure 13 and Figure 14 show the evolution of the posterior mean norm difference as the size $N$ of the dataset considered for the posterior distribution increases. For each value of $N$, 10 different datasets are used in order to quantify the variability of said error. The proportion of degenerate datasets is

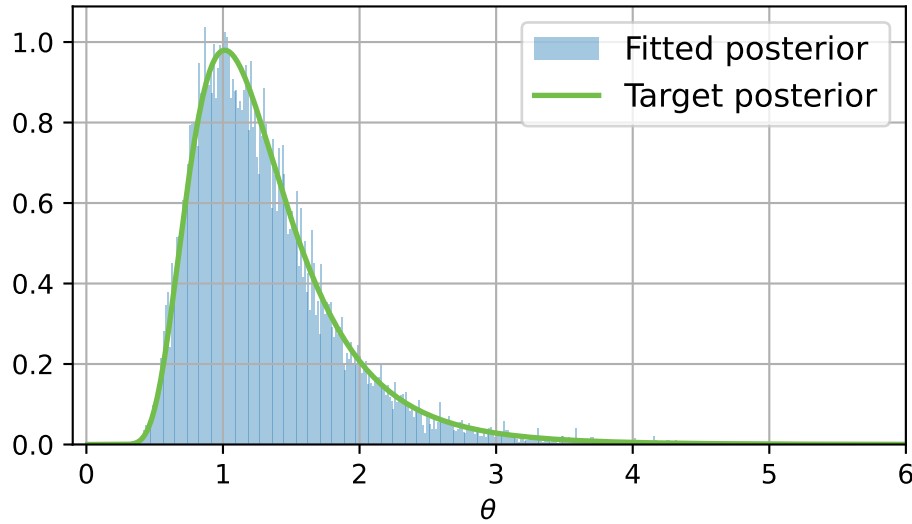

Figure 12: Histogram of the fitted posterior obtained from $5 \cdot 10^4$ samples, and density function of the target posterior.

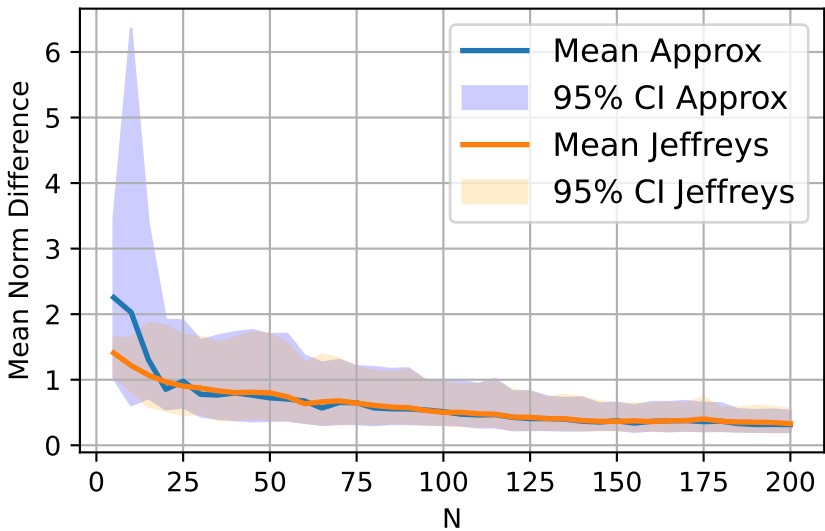

Figure 13: Mean norm difference as a function of the size $N$ of the dataset for the unconstrained fitted posterior and the Jeffreys posterior. For each value of $N$, 10 different datasets are considered from which we obtain 95% confidence intervals.

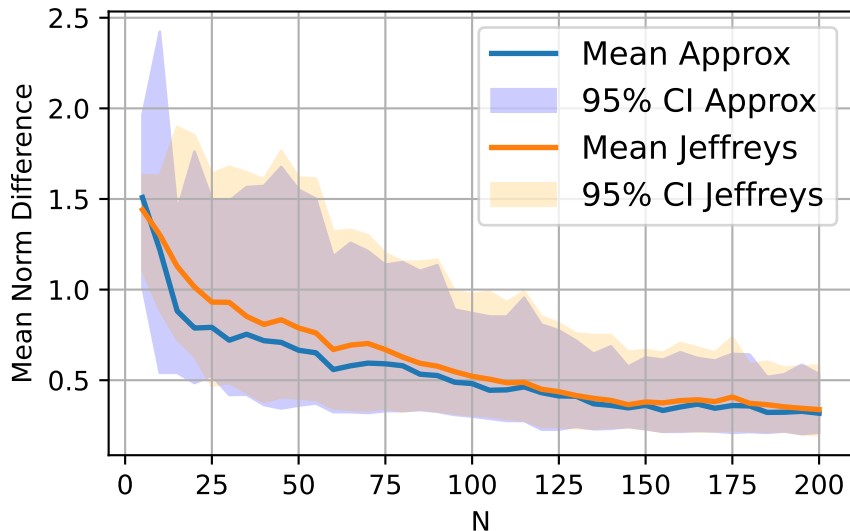

Figure 14: Mean norm difference as a function of the size $N$ of the dataset for the constrained fitted posterior and the Jeffreys posterior. For each value of $N$, 10 different datasets are considered from which we obtain 95% confidence intervals.

rather high when $N = 5$ or $N = 10$, the consequence is that the approximation tends to be more unstable. The main observation is that the error is decreasing in all cases when $N$ increases, also, the behaviour of the error for the fitted distributions on one hand, and the behaviour for the Jeffreys distribution on the other hand are quite similar in terms of mean value and confidence intervals.

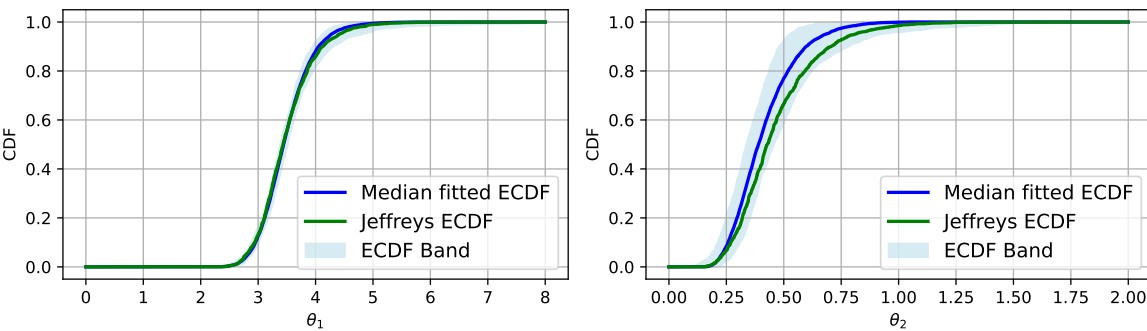

Figure 15: Empirical cumulative distribution functions for the unconstrained fitted posterior and the Jeffreys posterior using 5000 samples. The band is obtained by computing the ECDFs over 100 different seeds and monitoring the maximum and minimum ECDF values for each $\theta$.

Figure 15 and Figure 16 compare the empirical cumulative distribution functions of the fitted posterior and the Jeffreys posterior. In the unconstrained case, one can observe that the ECDFs are very close for $\theta_1$, whereas the variability is slightly higher for $\theta_2$ although still reasonable. When imposing a constraint on $\theta_2$, one remarks that the variability of the result is higher. The Jeffreys ECDF is contained in the band when $\theta_2$ is close to zero, but not when $\theta_2$ increases ($\theta_2 > 0.5$). This is coherent with the previous scatter histograms where the Jeffreys posterior on $\theta_2$ tends to have a heavier tail than the variational approximation.

Altogether, despite the stochastic nature of the developed algorithm, we consider that the result tends to be reasonably robust to the RNG seed for the optimization part, and robust to the dataset used for the posterior distribution for the MCMC part.

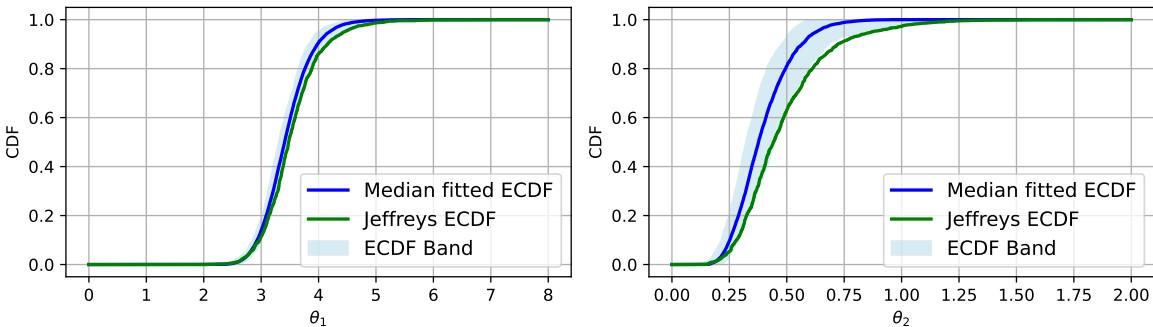

Figure 16: Empirical cumulative distribution functions for the constrained fitted posterior and the Jeffreys posterior using 5000 samples. The band is obtained by computing the ECDFs over 100 different seeds and monitoring the maximum and minimum ECDF values for each $\theta$.

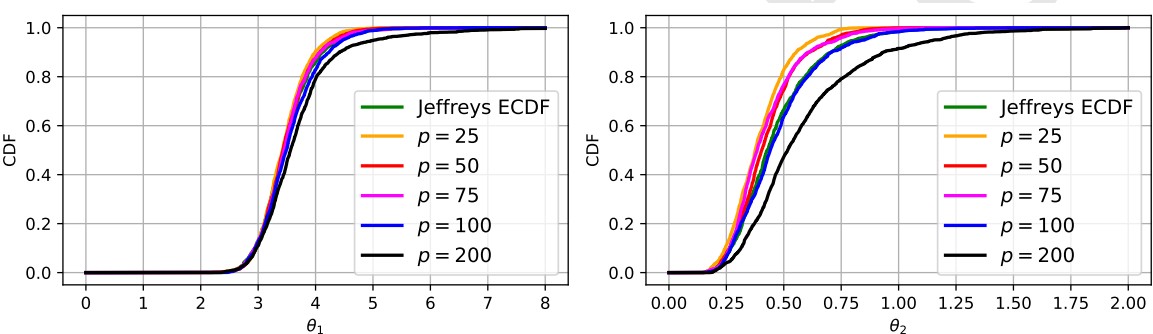

Figure 17: Empirical cumulative distribution functions for the constrained fitted posterior and the Jeffreys posterior using 5000 samples. The band is obtained by computing the ECDFs over 100 different seeds and monitoring the maximum and minimum ECDF values for each $\theta$.

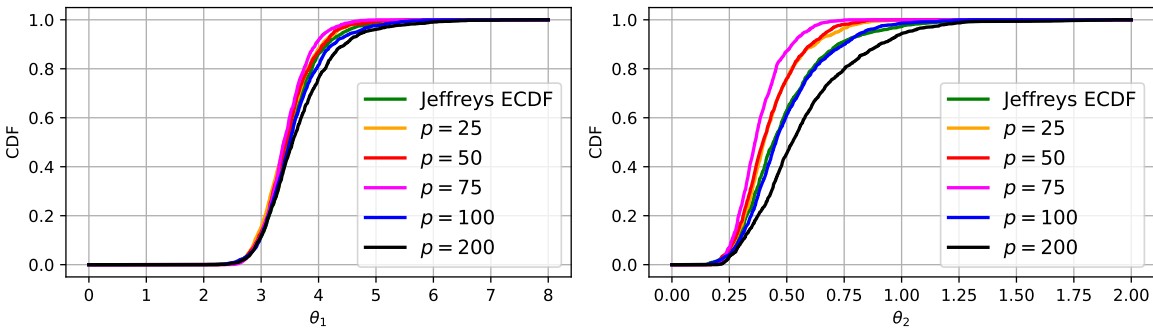

Figure 18: Empirical cumulative distribution functions for the constrained fitted posterior and the Jeffreys posterior using 5000 samples. The band is obtained by computing the ECDFs over 100 different seeds and monitoring the maximum and minimum ECDF values for each $\theta$.

Figure 17 and Figure 18 compare the empirical cumulative distribution functions of the fitted posterior and the Jeffreys posterior when several values for the latent space dimension $p$ are considered. We observe that in both the unconstrained case and the constrained case, the ECDFs are quite different for the $\theta_1$ component when $p$ varies, these differences are even more notable on $\theta_2$. We remark that the fitted distributions for $p = 100$ are the closest to the target Jeffreys distributions compared to lower values of $p$, but this is likely due to random chance, since when we keep increasing $p$ to 200, we obtain a worse approximation of the Jeffreys distributions. This last case is expected to be less stable due to the higher number of parameters to be fitted. The output of the algorithm is quite sensitive with respect to the choice of $p$ for the probit model, whereas for the multinomial model we noticed that this choice had little effect on the MMD values.

A possible explanation for this behavior can be obtained by looking at the approximation of the target prior given in reference Van Biesbroeck et al. (2025), which exhibits a correlation between $\theta_1$ and $\theta_2$. Thus, this allows us to numerically verify that even in the case where the prior is proper, the conditional variance of $\theta_2$ and the variance of $\theta_1$ are infinite due to the heavy tail in $\theta_2 \longrightarrow \infty$. The instability of the algorithm therefore seems to be due to the fact that it aims to approach a distribution of infinite variance.

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
