# OpenReview forum: "Variational inference for approximate objective priors using neural networks"
_Computo — Accepted by Computo_

### Review · Reviewer_XmgZ · 2025-03-27

**Summary Of Contributions:**

In this paper the authors develop a procedure to derive approximate (in certain sense) reference priors using a battery of numeric techniques, essentially based on variational inference. Such procedure, if really available, would be the "holy grail" of Bayesianism (at least the objective subset of it) as it would make feasible the application of the reference-priors paradigm without the need to solve their intrincate maths.

**Audience:**

Yes

**Broader Impact Concerns:**

Does not apply

**Claims And Evidence:**

No

**Requested Changes:**

The RP for the multinomial model is not the Dirichlet(0.5,...,0.5). There is a huge relevant literature on this model which is missed here (like the Bayesian Analysis paper by Berger, Bernardo and Sun). As the authors acknowldge this Dirichlet is the Jeffreys prior, but the Jeffreys prior is (by orders of magnitude) worst than the RP, particularly when the number of parameters increase (where the obtention of the RP becomes more complex and where having a numerical alternative would be more appealing). Is it the case that we are getting an approximation to the Jeffreys' prior?

The application of authors' procedure to the variance of an iid Normal model with known mean is paradigmatic of the limitations of the method. Obviously here the RP can be easily obtained but with their variational approach its approximation becomes a challenging numerical problem.

**Strengths And Weaknesses:**

My main criticism of the paper is that, what one finally seems to obtain is an approximation to an approximation to something which perhaps is close to the reference priors. Difficult to assess whether we are close to the original version of the prior or not. Still I think the research goes into the right direction and every small step towards an algorithmic version of the RP is welcome and, in my opinion, merits publication. Nevertheless, I would be more happy if the title, abstract and introduction are re-written in a way that more accurately reflect the actual contributions.

---

> ### Author Response · Authors · 2025-04-28
> **Answer to the review part 1**
>
> We thank the Reviewer for the helpful comments and suggestions.
> We took all the comments into account. We believe these edits have improved the
> clarity of the article. The github repository has been updated with a new version of the manuscript.
>
> The different corrections corresponding to the reviewer's remarks appear in green in the revised manuscript (in the html version but not in the pdf version).
>
> - My main criticism of the paper is that, what one finally seems to obtain is an approximation to an approximation to something which perhaps is close to the reference priors. Difficult to assess whether we are close to the original version of the prior or not.
>
> Answer : We have clarified the manuscript about what is our prior and what it approximates.
> More precisely, we admit and emphasize when we define the VA-RP that we aim to obtain the prior that yields the highest mutual information value, in the specific case where no ordering is imposed on the parameters, for a fixed value of N data samples in the likelihood function, and for a fixed parametrization determined by the chosen neural network.
>     Nevertheless, we have considered methods to assess that our priors are good approximations of the target priors (which are Jeffreys priors). When the solution is improper, we focus on comparing our posteriors with the target ones.
>     In the case of a proper target prior, the comparison is possible and the difference can be measured using metrics such as the maximum mean discrepancy, as presented in the multinomial model section.
>     This point has also been highlighted in the new version of the manuscript.
>
> - Nevertheless, I would be more happy if the title, abstract and introduction are re-written in a way that more accurately reflect the actual contributions.
>
> Answer : The title, abstract and introduction have been modified according to this remark.
>   We put more emphasis on the distinction between the reference priors and the priors maximizing the mutual information in multi-dimensional cases. We clarify the situations where those two definitions do not (hierarchical construction, ordering of the parameters) and do (when there are no ordering of the parameters) coincide. As our algorithm stands, only the particular case where no ordering is given can be tackled, hence, the target prior is either the Jeffreys prior when no constraint is imposed, or a modified Jeffreys prior when constraints are present. This is why for many occurrences, we replaced the term "reference prior" by "Jeffreys prior" when specifically talking about the target prior, and we replaced it by "objective priors" when no specific prior is designated.
>
> - The RP for the multinomial model is not the Dirichlet(0.5,...,0.5). There is a huge relevant literature on this model which is missed here (like the Bayesian Analysis paper by Berger, Bernardo and Sun). As the authors acknowledge this Dirichlet is the Jeffreys prior, but the Jeffreys prior is (by orders of magnitude) worst than the RP, particularly when the number of parameters increase (where the obtention of the RP becomes more complex and where having a numerical alternative would be more appealing). Is it the case that we are getting an approximation to the Jeffreys' prior?
>
> Answer : Indeed, there is consensus that other objective priors exist for the multinomial model that have more desirable properties than the Jeffreys prior. We now mention in the manuscript the work of Berger, Bernardo and Sun "Overall Objective Priors" and the work of Berger and Bernardo "Ordered group reference priors with application to the multinomial problem".
>     The reference prior suggested by the authors is the $m$-group reference prior where m equals the dimension of $\theta$, while the Jeffreys prior corresponds to $m=1$. While we acknowledge that the prior obtained for this model (which is close to a Dir(0.5,...,0.5)) is not the most appropriate prior to use in high-dimensional scenarios, we consider that it constitutes a good demonstration that our algorithm works beyond low dimensional models. We have clarified this difference between our target prior and the RP for this model in the new version of the manuscript.

---

> ### Author Response · Authors · 2025-04-28
> **Answer to the review part 2**
>
> - The application of authors' procedure to the variance of an iid Normal model with known mean is paradigmatic of the limitations of the method. Obviously here the RP can be easily obtained but with their variational approach its approximation becomes a challenging numerical problem.
>
> Answer : Indeed, the use of the algorithm is not necessary when the target distributions can be exactly computed, which is the case for this toy model. We added a remark stressing that this simple case must be considered as a sanity check, where one can compare the output of the algorithm to the true solution, notably when adding constraints. Conversely, in the probit model, we compare the output to a grid-based approximation of the Jeffreys prior, because no closed expression of the latter is available.
>
> Sincerely,
>
> The Authors

---

### Review · Reviewer_1JAZ · 2025-07-15

**Summary Of Contributions:**

This paper aims to compute particular objective prior distributions—specifically, reference priors, which may not be available in closed form. Since the definition of a reference prior is implicit, given as the solution to an optimization problem, the variational approach proposed by the authors is relevant. It also allows them to incorporate constraints to ensure that the prior is proper. They propose a sampler to obtain the posterior distribution when the prior is obtained via their variational approximation. Finally, three numerical examples are presented (two in the main text and one in the appendix). The goal is to compare the variational approximation of the prior with the exact expression when it is known. When the prior is improper, the comparison is instead made between the resulting posterior distributions.

**Audience:**

Yes

**Broader Impact Concerns:**

no concerns.

**Claims And Evidence:**

Yes

**Requested Changes:**

I may have missed something, but from my understanding, the authors are targeting the reference prior. Therefore, I would strongly suggest that they consistently use the term reference prior. A connection with the Jeffreys prior should be made, but the reader may be confused when the authors claim they are targeting the Jeffreys prior (e.g., in line 216 or line 418), while their optimization problem clearly corresponds to the reference prior framework. In Section 4.1, they should clarify that they are targeting the reference prior, which happens to coincide with the Jeffreys prior in this specific case. Since the prior and posterior distributions have closed-form expressions, it would be preferable in Figures 2 and 3 to use curves to represent the Jeffreys prior and the posterior distributions.

Section 3.2 is hard to read: additional notations are introduced without being defined (e.g., $\tilde{I}$, $\tilde{O}_D$). This section is particularly difficult to follow unless the reader consults Appendix 6.1 simultaneously. The authors should choose either to simplify this section and refer to a self-contained Appendix 6.1, or to add more detail here to ensure that the section is understandable on its own.

**Strengths And Weaknesses:**

The approach is interesting and appears efficient based on the test cases. It is also quite comprehensive, as it addresses both posterior sampling and a way to enforce the properness of the prior distribution.
However, I found the exposition very confusing, as the authors use the terms reference prior and Jeffreys prior interchangeably. While the two coincide in the specific case of a one-dimensional parameter and under asymptotic conditions, this is not always true, and the methods for deriving a reference prior or a Jeffreys prior are inherently different.
The numerical study could be strengthened by incorporating a sensitivity analysis with respect to the neural network settings (e.g., varying the number of latent dimensions).

---

> ### Author Response · Authors · 2025-08-26
> **Answer to the review**
>
> We thank the Reviewer for the helpful comments and suggestions.
> We took all the comments into account. We believe these edits have improved the
> clarity of the article. The Github repository has been updated with a new version of the manuscript. The different corrections corresponding to the reviewer's remarks appear in orange in the revised manuscript but only in the html version, not the pdf version. The corrections in green follow from the first review.
>
> - The numerical study could be strengthened by incorporating a sensitivity analysis with respect to the neural network settings (e.g., varying the number of latent dimensions).
>
> Answer : We incorporated in the numerical experiments a sensitivity analysis with respect to the dimension of the latent space in both the multinomial model and the probit model.
>
> - I may have missed something, but from my understanding, the authors are targeting the reference prior. Therefore, I would strongly suggest that they consistently use the term reference prior. A connection with the Jeffreys prior should be made, but the reader may be confused when the authors claim they are targeting the Jeffreys prior (e.g., in line 216 or line 418), while their optimization problem clearly corresponds to the reference prior framework.
>
> Answer : We added a few paragraphs in Section 2 in order to clarify the differences and the similarities between every prior invoked in said section. The Jeffreys prior is a special case of reference prior when no ordering on the parameters is imposed for the multi-dimensional case.  Even in this simpler situation, the maximization of the mutual information remains a difficult problem. Thus, we only address this particular case in our work, this is why the Jeffreys prior is indeed the target prior.
>
> - In Section 4.1, they should clarify that they are targeting the reference prior, which happens to coincide with the Jeffreys prior in this specific case. Since the prior and posterior distributions have closed-form expressions, it would be preferable in Figures 2 and 3 to use curves to represent the Jeffreys prior and the posterior distributions.
>
> Answer : We believe that our modifications have clarified the differences between the reference priors when an ordering is imposed on the parameters and the reference priors when no such ordering is set. The latter corresponds to the Jeffreys prior, which is our target. As for Figures 2 and 3, to illustrate the target prior and posterior distributions, we replaced the marginal histograms obtained with random samples by the marginal densities functions since they can be computed exactly in this case. We also replaced in Figure 9, appendix 6.2, the histogram of the target distribution by its density function for the unconstrained posterior calculation since the same remark applies.
>
> - Section 3.2 is hard to read: additional notations are introduced without being defined (e.g., $\tilde{I}$, $\tilde{O}_D$). This section is particularly difficult to follow unless the reader consults Appendix 6.1 simultaneously. The authors should choose either to simplify this section and refer to a self-contained Appendix 6.1, or to add more detail here to ensure that the section is understandable on its own.
>
> Answer : We have clarified section 3.2 by changing or detailing some notations in order to make this part self-contained. The derivations made in the appendix are not mandatory to understand the general principle of the algorithm.
>
> Sincerely,
>
> The Authors

---

### Review · Reviewer_pJ1q · 2025-11-26

**Summary Of Contributions:**

The paper addresses the problem of computing reference prior distributions that are often intractable and can be marred in practice by their computational cost. Since reference priors are defined implicitly as maximizers of a mutual information based criterion, the authors propose to approximate them using variational inference, restricting attention to a parametric family of priors represented implicitly via neural networks. The authors tackle cases where the reference prior coincides with the Jeffreys prior.

Compared to previous work, the proposed method is extended to a generalized mutual information based on $f$-divergences (with a focus on $\alpha$-divergences in the numerical study), and allows the use to enforce linear constraints on the prior in order to obtain proper  priors.

To make the approach usable in Bayesian inference, the authors also propose a way to sample from the posterior by operating in the latent space. The approach is tested on several benchmark models (multinomial model, probit model, and a Gaussian variance model in the appendix).

**Audience:**

Yes

**Claims And Evidence:**

Yes

**Requested Changes:**

I do not request changes.

**Strengths And Weaknesses:**

The paper is well written and mathematically rigorous, with clear motivation and structure. Up to my knowledge, the application of modern variational framework to objective prior computation is novel. The proposed method is conceptually sound and effectively addresses key issues for an end-to-end Bayesian approach. In particular, with the contribution to enforce constraints and the derivation of a sampling method (via the latent space), the method is of practical value. The experiments confirm the efficiency of the algorithm, including for instance meaningful comparisons to known analytical Jeffreys priors.

In conclusion, this paper is a good theoretical and methodological contribution, providing a bridge between classical reference/Jeffreys priors and modern machine learning tools. I therefore recommend it for publication.

---

### Comment · Action_Editor_1CX8 · 2025-02-27
**HTML version  of the paper**

The HTML version of the manuscript is available at https://nilsbaillie.github.io/Computo_VA-RP/

---

### Comment · Action_Editor_1CX8 · 2025-07-29
**Rebuttal: discussion and revision period**

Dear authors,

We have received two reports for your submission to Computo.

By default, a period of 6 weeks is allowed for discussion with the referees before they issue a final opinion. During this period, you can make any changes to your submission that you feel are necessary and that you are able to make.  I note that you did not wait for our request to respond to the first reviewer, which is a good initiative. You can now respond to the second reviewer and make any changes to your submission that you deem necessary.

At the end of this period (scheduled for August 27), the reviewers will have about two weeks to make their decision, ranging from final acceptance to more substantial requests for modification. Given that it is summer, if you need an extra week or two, please let me know as soon as possible.

Best regards

---

### Comment · Editors_In_Chief · 2025-10-06
**End of rebuttal period - your recommendation is needed**

Dear reviewer,

The rebuttal period has now ended and the authors have responded and amended their manuscript accordingly: based on this, could you make your official recommendation via OpenReview (reject, accept, resubmit)?

Many thanks

Julien C., EiC

---

### Comment · Editors_In_Chief · 2025-10-27
**Processing time for your submission**

Dear authors,

I would like to apologize for the delay in reviewing your article: an initial reviewer committed to reviewing it but ultimately did not respond to our request. A second reviewer did their job properly and responded to your resubmission. A third reviewer made comments but was unable to provide an assessment of your resubmission.

I have therefore requested a new, simplified external evaluation to help me make a final decision.

Thank you for your patience.

---

### Comment · Editors_In_Chief · 2025-11-26
**The production phase of your article is about to start**

Dear author,

Congratulations on the acceptance! Before the production phase of your paper starts, we need you to perform a couple of steps.

1. Can you please check that your affiliation and metadata are correctly filled in in the header of your article (name, affiliation, and url of each author).
2. Please run the CI and fix any remaining issues with the build.
3. Finally, transfer the ownership of the github repository to computo. See the [Authors guidelines](https://computo-journal.org/site/guidelines-authors.html#sec-acceptance) for a small description on how to perform the ownership transfer. In order to do so, you have been temporarily added to the computo organization.

Many thanks

---

### Note · Reviewer_1JAZ · 2025-09-19

**Comment:**

My comments have been properly taken into account and this version of the paper is ready for publication in computo.

**Audience:**

Yes

**Claims And Evidence:**

Yes

**Decision Recommendation:**

Accept

---

### Note · Action_Editor_1CX8 · 2025-11-26

**Comment:**

The comments raised by the first two reviewers were taken into account, even though the first reviewer did not confirm them during a second review.

A third opinion definitively confirmed the relevance of the submission to Computo's scope and the accuracy of the analyses. I recommend acceptance.

**Audience:**

Yes

**Claims And Evidence:**

Yes

**Decision Recommendation:**

Accept

---

### Decision · Action_Editor_1CX8 · 2025-11-26

**Recommendation:** Accept as is

**Comment:**

See reviewers' reports

**Audience:**

This paper is relevant for computo's readership.

**Claims And Evidence:**

yes

---

> ### Decision · Editors_In_Chief · 2025-11-26
>
> I approve the AE's decision.